# ClusterFusion: Expanding Operator Fusion Scope for LLM Inference via Cluster-Level Collective Primitive

**Xinhao Luo**[1,2]    **Zihan Liu**[1,2*]    **Yangjie Zhou**[3*]    **Shihan Fang**[1]    **Ziyu Huang**[1,2]    **Yu Feng**[1,2]
**Chen Zhang**[1]    **Shixuan Sun**[1]    **Zhenzhe Zheng**[1]    **Jingwen Leng**[1,2]    **Minyi Guo**[1,2]

[1] Shanghai Jiao Tong University    [2] Shanghai Qi Zhi Institute    [3] National University of Singapore

{lxh666, altair.liu, fang-account, huang_ziyu}@sjtu.edu.cn
{y-feng, chenzhang.sjtu, sunshixuan, zhengzhenzhe}@sjtu.edu.cn
yj_zhou@nus.edu.sg, leng-jw@cs.sjtu.edu.cn, guo-my@cs.sjtu.edu.cn

## Abstract

Large language model (LLM) decoding suffers from high latency due to fragmented execution across operators and heavy reliance on off-chip memory for data exchange and reduction. This execution model limits opportunities for fusion and incurs significant memory traffic and kernel launch overhead. While modern architectures such as NVIDIA Hopper provide distributed shared memory and low-latency intra-cluster interconnects, they expose only low-level data movement instructions, lacking structured abstractions for collective on-chip communication. To bridge this software-hardware gap, we introduce two cluster-level communication primitives, `ClusterReduce` and `ClusterGather`, which abstract common communication patterns and enable structured, high-speed data exchange and reduction between thread blocks within a cluster, allowing intermediate results to be on-chip without involving off-chip memory. Building on these abstractions, we design `ClusterFusion`, an execution framework that schedules communication and computation jointly to expand operator fusion scope by composing decoding stages such as *QKV Projection*, *Attention*, and *Output Projection* into a single fused kernels. Evaluations on H100 GPUs show that `ClusterFusion` outperforms state-of-the-art inference frameworks by $1.61\times$ on average in end-to-end latency across different models and configurations. The source code is available at https://github.com/xinhao-luo/ClusterFusion.

## 1  Introduction

Large language models (LLMs) have become a cornerstone of modern artificial intelligence systems. Their applications span natural language processing [49, 25], code generation [17, 42], and mathematical reasoning [45, 54]. LLM inference typically involves two stages: a prefilling phase that encodes the input prompt and a decoding phase that generates output tokens auto-regressively. As sequence length increase and model sizes grow, the decoding phase dominates overall inference latency, making it the primary bottleneck in real-time LLM applications.

To accelerate decoding, recent research has explored various optimizations [10, 19, 26, 29, 15, 16, 13, 14, 26]. A growing body of research focuses on optimizing execution dataflow [10, 19, 55, 22], which refers to the organization of computation and communication across the parallel and memory hierarchy of modern GPUs [34]. Thread blocks serve as the fundamental execution units, each responsible for processing a portion of the data and typically assigned to different hardware unit such

---

*Corresponding authors.

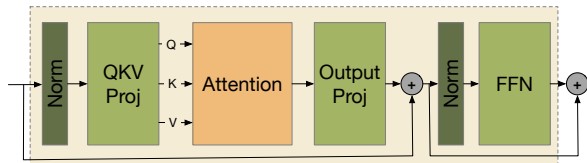

Figure 1: Typical Transformer Block as the fundamental component of the modern LLMs.

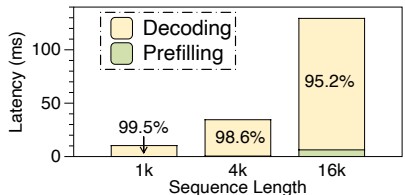

Figure 2: Latency comparison of prefilling and decoding for 256 tokens.

as streaming multiprocessor (SM). However, most existing systems treat thread blocks as independent execution units. Inter-block dependencies are resolved by materializing intermediate results to global memory, resulting in frequent off-chip transfers, redundant synchronization, and limited operator fusion scope [63, 53].

Recent GPU architectures offer new opportunities to address these limitations. NVIDIA Hopper introduces thread block clusters and distributed shared memory (DSMEM), which enable direct on-chip communication among blocks within the same cluster [33]. Despite this potential, two major challenges remain. First, current architectures expose only low-level data movement instructions without providing high-level structured communication abstractions. Second, as our analysis in Sec. 2.3 shows, DSMEM performance is highly sensitive to cluster configuration. These factors make it challenging to integrate DSMEM into real-world LLM systems.

To address these limitations, we propose two cluster-level collective primitives: `ClusterReduce` and `ClusterGather`, which abstract common communication patterns such as reduction and aggregation. These primitives enable structured intra-cluster coordination, allowing intermediate results to be shared and combined on-chip without global memory access. Built upon these primitives, our key insight is to treat each thread block cluster as a fundamental parallel unit, using cluster-level collective communication primitives to resolve inter-block dependencies efficiently. Guided by the key insight, we propose the cluster-centric dataflow and develop `ClusterFusion`, an execution framework that jointly schedules computation and communication to expand operator fusion scope.

Evaluation on NVIDIA H100 GPUs shows that `ClusterFusion` achieves $1.61\times$ speedup on average in end-to-end latency compared to state-of-the-art frameworks. These performance gains hold across diverse model architectures (e.g., Llama [51], DeepSeek [11]) and configurations, demonstrating the generality and effectiveness of our approach.

In summary, we have made the following contributions:

• We analyze the communication patterns and fusion scope in existing LLM decoding workflows, identifying fragmented kernel execution and off-chip synchronization as key barriers to efficient decoding fusion. We further profile the DSMEM mechanism on NVIDIA Hopper GPUs, revealing its potential to support low-latency inter-block communication and reduce off-chip memory dependency.

• We propose two cluster-level collective primitives, `ClusterReduce` and `ClusterGather`, to support structured inter-block collective communication. These primitives abstract reduction and aggregation over DSMEM and enable efficient coordination between thread blocks.

• We develop `ClusterFusion`, an execution framework that expands operator fusion via our proposed primitives. `ClusterFusion` integrates structured intra-cluster communication into the cluster-centric dataflow, fusing *QKV Projection*, *Attention* and *Output Projection*. This enables coordinated computation and communication without off-chip memory traffic and outperforms SOTA frameworks.

## 2 Background

### 2.1 LLM Inference Workflow and Bottlenecks

LLMs are commonly built on Transformer-based decoder-only architectures [52]. As shown in Fig. 1, each Transformer Block comprises *QKV Projection*, *Attention*, *Output Projection* and a feed-forward network (FFN). For an input sequence $X$, each *Attention* head computes projections $Q_i = XW_{Q_i}$, $K_i = XW_{K_i}$, $V_i = XW_{V_i}$, followed by scaled dot-product *Attention* and *Output*

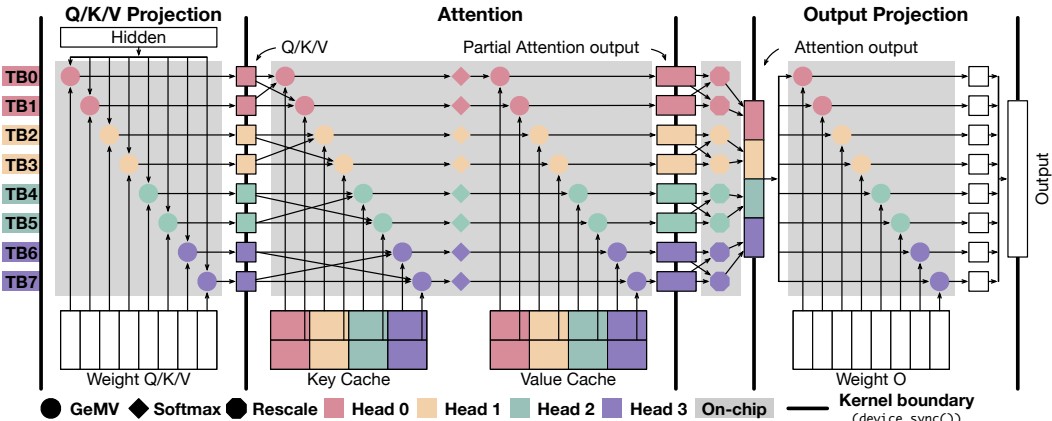

Figure 3: Existing *QKV Projection*, *Attention*, and *Output Projection* dataflow graph.

*Projection*:

$$Z = \text{Concat}\left(\text{Softmax}\left(\frac{Q_1 K_1^\top}{\sqrt{d_k}}\right) V_1, \ldots, \text{Softmax}\left(\frac{Q_h K_h^\top}{\sqrt{d_k}}\right) V_h\right) W_O \tag{1}$$

Here, $d_k$ is the dimension of each head's query (Q) and key (K). The FFN module applies three linear layers with a non-linear activation in between, formulated as:

$$\text{FFN}(Z) = W_3 \left(\sigma(W_1 Z) \odot W_2 Z\right) \tag{2}$$

where $\sigma$ is a non-linear activation (e.g., GELU), and $\odot$ denotes element-wise multiplication.

During inference, the model proceeds in two distinct stages: prefilling and decoding. In the decoding stage, the model generates tokens one at a time in an auto-regressive manner, reusing the KV cache while appending new entries. This sequential decoding pattern inherently limits parallelism and dominates inference latency at longer context lengths. In Fig. 2, the percentages represent the proportion of latency on decoding stage for different sequence lengths. Decoding accounting for over 95% of the total latency for a 256 tokens sequence generation, as measured using SGLang [58], a state-of-the-art LLM serving framework. This makes decoding the dominant computational bottleneck during inference and a natural target for system-level performance optimization.

## 2.2 Existing Communication Patterns and Fusion Scope

Numerous studies have proposed various techniques [9, 8, 44, 10, 19, 55, 57] to improve the Transformer performance. Among them, increasing attention has been given to the design of execution dataflow, which refers to the structured organization of computation stages and their associated data movement, including partitioning, scheduling, and communication, over the parallel execution model and memory hierarchy [34]. In decoding workloads, the dataflow plays a central role in determining end-to-end latency by governing operator scheduling and intermediate data exchange.

In existing GPU-based decoding dataflows, thread blocks commonly serve as the fundamental unit of parallel execution and scheduling. Fig. 3 illustrates the dataflow of Llama2-7B [51] decoding phase in SGLang [58], covering three stages: *QKV Projection*, *Attention*, and *Output Projection*. Within each kernel, thread blocks are assigned to individual *Attention* heads and operate on disjoint tiles of the hidden dimension and KV sequence. In the *QKV Projection* stage, each thread block performs a linear transformation on its assigned hidden states to produce local Q, K, and V vectors. These outputs are written to global memory. The *Attention* stage is implemented with FlashDecoding[10, 19, 55], where each block computes a partial *Attention* result using its corresponding Q and a segmented KV cache first. A separate rescaling kernel then aggregates these partial results across blocks. The final *Output Projection* is executed block-wise on the aggregated *Attention* output. Existing decoding dataflows exhibit two forms of inter-block communication: exchanging intermediate data (e.g., Q/K/V vectors) needed by multiple blocks, and reducing partial results across blocks (e.g., Attention output). These dependencies are resolved via off-chip memory and explicit kernel boundaries, leading to global synchronization barriers and hindering operator fusion [63, 53].

This block-isolated execution structure introduces launch overhead, off-chip memory round-trips, and global synchronization barriers between kernels. Due to the absence of structured communication across thread blocks, intermediate results must be materialized to global memory and reloaded by subsequent kernels, limiting opportunities for effective on-chip data reuse and broader fusion. To overcome these limitations, we require an execution mechanism that enables collective scheduling and communication across thread blocks.

## 2.3 Cluster-Level Opportunities and Challenges

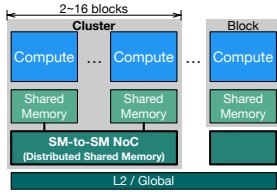

Figure 4: NVIDIA Hopper architecture.

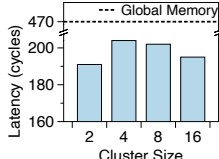 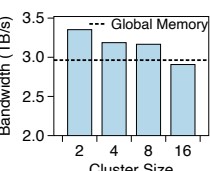 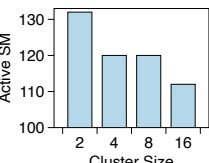

Figure 5: SM-to-SM access latency (left), bandwidth (middle), and number of active SMs (right) for varying cluster size on H100 GPU.

The previous section shows that the mainstream methods resolve inter-block data dependencies via global memory synchronization. However, recent architectures, such as NVIDIA Hopper [33] shown in Fig. 4, can group a set of thread blocks into a cluster. Within each cluster, thread blocks can share data directly through a high-speed SM-to-SM Network-on-Chip (NOC), also referred to as DSMEM, thereby avoiding costly global memory accesses and enabling efficient intra-cluster communication.

To understand the opportunities and challenges of this mechanism, we profile DSMEM on an NVIDIA H100 GPU by varying the cluster size from 1 to 16, which is the maximum supported by Hopper hardware. As shown in Fig. 5 left, SM-to-SM access latency improves significantly with small cluster sizes. When the cluster size is 2, the average latency reaches 190 cycles, which is substantially lower than global memory latency (exceeding 470 cycles). This demonstrates the potential of DSMEM for low-latency on-chip communication.

However, this benefit comes with notable trade-offs. As the cluster size increases, the available communication bandwidth decreases, slightly lagging behind the global memory bandwidth when the cluster size reaches 16 (2.90 TB/s vs. 2.96 TB/s) due to crossbar architecture [23, 3]. Additionally, the number of active SMs is reduced due to hardware constraints. These effects limit the scalability of cluster-based execution and necessitate careful configuration to balance communication efficiency and overall parallelism.

Beyond hardware-level trade-offs, there are software-level challenges as well. NVIDIA currently exposes DSMEM and thread block cluster functionality only through low-level PTX instructions, which only support basic, peer-to-peer data movement between thread blocks [33]. This low-level interface presents significant challenges for expressing reusable synchronization and communication patterns in practical scenarios, which results in a steep programming barrier and leaves developers without clear guidance on how to effectively apply these capabilities.

## 3 ClusterFusion: Execution Framework with Cluster-Centric Dataflow

This section presents the design of `ClusterFusion`. We begin by introducing cluster-level collective primitives that enable structured data reduction and aggregation across thread blocks. These primitives form the foundation for the cluster-centric dataflow that fuses *QKV Projection*, *Attention* and *Output Projection*. `ClusterFusion` builds on this to achieve high-performance LLM decoding with expanded operator fusion scope.

### 3.1 Cluster-Level Collective Primitives

To explore hardware-level trade-offs and overcome software-level challenge, we introduce two cluster-level collective primitives that abstract common communication patterns. The key insight is to treat a thread block cluster as a fully connected logical network, where blocks participate in structured collective operations. Inspired by communication primitives in distributed systems [48, 4], we design

---

**Algorithm 1** ClusterReduce over DSMEM - Thread Block view

---

**Require:** A cluster of $N = 2^k$ thread blocks ($k \leq 4$), each with rank $b \in [0, N-1]$ and a shared memory buffer $\mathbf{D}_b$ containing local data that needs to be reduced together with data from other thread blocks in the same cluster. A reduction operator $\oplus$ (e.g., sum or max).
1: Allocate shared memory buffer $\mathbf{B}_b$ with the same size of $\mathbf{D}_b$        ▷ Used as a temporary buffer.
2: stride ← 1.
3: **while** stride < $N$ **do**
4:     block rank send_to ← $(b + \text{stride}) \bmod N$.
5:     block rank recv_from ← $(b - \text{stride} + N) \bmod N$.
6:     Send $\mathbf{D}_b$ to $\mathbf{B}_{\text{send\_to}}$ of block send_to via DSMEM.
7:     Receive $\mathbf{D}_{\text{recv\_from}}$ from block recv_from into $\mathbf{B}_b$ via DSMEM.
8:     Wait for the arrival of $\mathbf{D}_{\text{recv\_from}}$.
9:     $\mathbf{D}_b \leftarrow \mathbf{D}_b \oplus \mathbf{B}_b$        ▷ Aggregate partial result using a reduction operator.
10:     stride ← stride × 2        ▷ Exponential stride progression.
11: **end while**
12: Return $\mathbf{D}_b$.

---

---

**Algorithm 2** ClusterGather over DSMEM - Thread Block view

---

**Require:** A cluster of $N = 2^k$ thread blocks ($k \leq 4$), each with rank $b \in [0, N-1]$ and a shared memory buffer $\mathbf{D}_b$ of size $N \times$ size. The first segment $\mathbf{D}_b[0 : \text{size}]$ contains the local data that needs to be gathered together with data from other thread blocks in the same cluster.
1: stride ← 1.
2: **while** stride < $N$ **do**
3:     block rank send_to ← $(b + \text{stride}) \bmod N$.
4:     block rank recv_from ← $(b - \text{stride} + N) \bmod N$.
5:     Send $\mathbf{D}_b[0 : \text{size} \times \text{stride}]$ to $\mathbf{D}_{\text{send\_to}}[\text{stride} \times \text{size} : 2 \times \text{stride} \times \text{size}]$ of block send_to via DSMEM.
6:     Receive $\mathbf{D}_{\text{recv\_from}}[0 : \text{size} \times \text{stride}]$ from block recv_from into $\mathbf{D}_b[\text{stride} \times \text{size} : 2 \times \text{stride} \times \text{size}]$ via DSMEM.
7:     Wait for the arrival of $\mathbf{D}_{\text{recv\_from}}[0 : \text{size} \times \text{stride}]$.
8:     stride ← stride × 2        ▷ Exponential stride progression
9: **end while**
10: Return $\mathbf{D}_b$.

---

two cluster-level primitives: ClusterReduce, which reduces data across blocks using associative operators such as sum or max, and ClusterGather, which replicates local data from each block to all others for data sharing as shown in Fig. 6.

As shown in Alg. 1 and Alg. 2, both ClusterReduce and ClusterGather adopt a binary-tree pattern across $log_2 N$ rounds, where $N$ is the cluster size. In each round, the communication stride doubles, and each block exchanges data with a peer whose index is offset by the current stride. ClusterReduce performs element-wise reductions while keeping the message size constant, whereas ClusterGather progressively accumulates remote data by doubling the message size in each round. This shared structure facilitates uniform implementation and hardware tuning, enabling efficient cluster-level synchronization and data sharing.



Figure 6: Illustration of cluster-level collective communication primitives: ClusterReduce and ClusterGather.

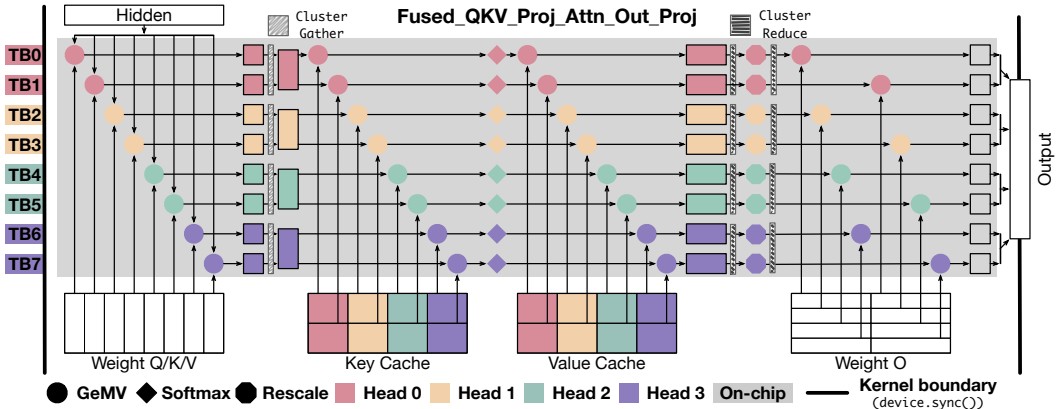

Figure 7: Cluster-centric fused *QKV Projection*, *Attention* and *Output Projection* dataflow graph.

## 3.2 Cluster-Centric Dataflow Design with Primitives

Building upon the proposed cluster-level primitives, we now illustrate how they are employed to construct a cluster-centric dataflow with expanded operator fusion scope. The core idea is to treat the **thread block cluster** as a cooperative execution and scheduling unit. Data-dependent dimensions are kept within each cluster to resolve inter-block data dependencies using cluster-level collective primitives which avoid off-chip data exchange, while data-independent dimensions are distributed across clusters. This organization aligns computation with memory locality and enables seamless kernel fusion by allowing intermediate results to be reused entirely within on-chip memory.

In the decoding stage, the dataflow for the Projection and *Attention* modules is parallelized over multiple dimensions. Specifically, the Projection is parallel along the number of heads and head dimension, while the *Attention* is parallel over the number of heads and the KV cache sequence length. Among these dimensions, thread blocks that compute different partitions of the head dimension and the KV cache sequence length exhibit inter-block data dependencies, since each block computes either a partial result that requires reduction or a segment result that needs to be gathered to form the final output. These thread blocks can be grouped into a cluster to resolve data dependencies on-chip by using cluster-level collective primitives. Since *Attention* heads are independent across all three modules, each cluster is accordingly mapped to a single head. As shown in Fig. 7, under this design, the intermediate results produced by the *QKV Projection* naturally remain on chip and are directly reused by the *Attention* module. Likewise, the output of the *Attention* module stays on chip and is immediately consumed by the *Output Projection*, enabling seamless data reuse across three modules.

By leveraging cluster-level communication primitives, we implement the fused *QKV Projection*, *Attention* and *Output Projection* dataflow. As illustrated in Alg. 5, the dataflow is parallel in the number of heads. Each head corresponds to a cluster of $N = 2^k$ thread blocks ($k \leq 4$), where each block is assigned a rank $b \in [0, N-1]$. Within each cluster, thread blocks respectively partition the head dimension in *QKV Projection*, the KV cache token dimension in *Attention*, and the output dimension in *Output Projection*. For the whole dataflow, each thread block processes the entire input hidden states and computes the corresponding output tile $O_b$ after the *Output Projection*. In this algorithm, $B$ denotes the batch size, $D$ the input hidden dimension, $H$ the total head dimension, and $h$, $s$ and $d$ represent the partitioned sizes of the head dimension, sequence length and the output dimension, per thread block, respectively.

According to our cluster-centric dataflow design principle, we also propose several dataflow variants, which are presented in the Appendix B. To evaluate these variants, we conduct a quantitative analysis of DSMEM traffic [47, 59]. We begin by analyzing the DSMEM memory traffic incurred by the `ClusterReduce` and `ClusterGather` primitives as follows:

$$Traffic_{Reduce}(size, N) = size \times \log_2 N \times N, \quad Traffic_{Gather}(size, N) = size \times \left(2^{\log_2 \frac{N}{2}+1} - 1\right) \times N$$

Here, $Traffic_{Reduce}$ and $Traffic_{Gather}$ denote the DSMEM traffic for `ClusterReduce` and `ClusterGather`, respectively. The variable `size` represents the size of the shared memory buffer $D_b$ in Alg. 1, as well as the initial segment size in Alg. 2. Based on this analytical model, we estimate

**Algorithm 3** Fused *QKV Projection*, *Attention* and *Output Projection* Dataflow - Thread Block View

---

**Require:** Input hidden states $\mathbf{H}_b \in \mathbb{R}^{1 \times D}$, *QKV Projection* weight $\mathbf{W}_b^{QKV} \in \mathbb{R}^{D \times 3h}$, *Output Projection* weight $\mathbf{W}_b^O \in \mathbb{R}^{H \times d}$, and KV cache $\mathbf{K}_b^{\text{cache}}, \mathbf{V}_b^{\text{cache}} \in \mathbb{R}^{s \times H}$ in global memory.

1: Allocate shared memory buffers: $\mathbf{Q}_b, \mathbf{K}_b, \mathbf{V}_b \in \mathbb{R}^{1 \times h}$, and $S_{\text{sum}}, S_{\text{max}}$ (softmax statistics).
2: Compute segment results of *QKV Projection*: $\mathbf{Q}_b, \mathbf{K}_b, \mathbf{V}_b \leftarrow \mathbf{H}_b \times \mathbf{W}_b^{QKV}$.
3: Obatin the complete QKV: $(\mathbf{Q}_b, \mathbf{K}_b, \mathbf{V}_b) \leftarrow \texttt{ClusterGather}(\mathbf{Q}_b, \mathbf{K}_b, \mathbf{V}_b)$.
4: Compute partial result of *Attention* similar to the FlashDecoding dataflow:
    Compute $\mathbf{S}_b \leftarrow \exp(\mathbf{Q}_b \times (\mathbf{K}_b^{\text{cache}}, \mathbf{K}_b)^T)$, obtain local $S_{\text{sum}}, S_{\text{max}}$.
    And store $S_{\text{max}}$ in register $Reg_{\text{max}}$.           ▷ softmax statistics.
    Compute $\mathbf{A}_b \leftarrow \mathbf{S}_b \times (\mathbf{V}_b^{\text{cache}}, \mathbf{V}_b)$   ▷ $\mathbf{A}_b$ reuse the shared memory space of $\mathbf{Q}_b$
5: Obtain the complete softmax statistics $S_{\text{sum}}$ and $S_{\text{max}}$:

$$S_{\text{sum}} \leftarrow \texttt{ClusterReduce}(S_{\text{sum}}, \text{sum}), \quad S_{\text{max}} \leftarrow \texttt{ClusterReduce}(S_{\text{max}}, \text{max})$$

6: Rescale *Attention* output according to online softmax:

$$\mathbf{A}_b \leftarrow \frac{\mathbf{A}_b \cdot \exp(Reg_{\text{max}} - S_{\text{max}})}{S_{\text{sum}}}$$

7: Obatin the complete *Attention* output: $\mathbf{A}_b \leftarrow \texttt{ClusterReduce}(\mathbf{A}_b, \text{sum})$.
8: Compute the result of the *Output Projection* and write it to global memory using `atomicAdd`:

$$\mathbf{O}_b \leftarrow \mathbf{A}_b \times \mathbf{W}_b^O$$

9: **return** $\mathbf{O}_b$.

---

the total DSMEM memory traffic of the dataflow used in Alg. 5, which includes one `ClusterGather` and two `ClusterReduce` operations:

$$Traffic_{Total} = Traffic_{Reduce}(3h, N) + Traffic_{Gather}(H, N)$$

We omit the traffic generated by the softmax statistics, as it involves only two floats and is negligible compared to tensor. According to the analysis shown in Appendix B, this dataflow yields the lowest DSMEM traffic and achieves better performance compared to other dataflow variants.

The proposed cluster-centric dataflow design principle can be naturally generalized to other operators, including DeepSeek MLA [11]. The corresponding algorithms are provided in the Appendix B. We implement an end-to-end execution framework, `ClusterFusion`, based on the cluster-centric dataflow, which incorporates the aforementioned fused *QKV Projection*, Attention, and *Output Projection* modules. For other components such as FFN and RMSNorm, we adopt optimized implementations consistent with those in existing frameworks such as CUTLASS [36] and Flashinfer [55].

## 4 Evaluation

**Experimental Setup** We evaluate `ClusterFusion` on an NVIDIA H100 SXM5 80GB GPU [33]. For the end-to-end evaluation, all inputs are in FP16 precision, and the context length varies from 1K to 16K. We set the batch size to 1; results of multi batch are presented in the Appendix C. All experiments are conducted using PyTorch 2.5.1 [40] and CUDA 12.4 [34].

**Baselines** We compare `ClusterFusion` with four state-of-the-art LLM inference frameworks: SGLang 0.4.3.post2 [58], vLLM 0.6.4.post1 [24], TensorRT-LLM 0.18.0 [39], and MLC-LLM 0.20.dev0 [32]. For all baselines, we use the recommended configurations from their official documentation, including backend kernels from libraries such as CUTLASS [36], FlashAttention [9, 8, 44, 57], and FlashInfer [55], or generated by Triton [50] and TVM [7]. All frameworks enable CUDA Graph [35] and Torch.compile [5].

**Models** `ClusterFusion` is evaluated on two representative LLMs: Llama2-7B [51] and DeepSeek-V2-Lite [11], both based on the Transformer architecture. Llama2-7B adopts standard Multi-Head Attention (MHA) mechanism, while DeepSeek-V2-Lite employs Multi-head Latent Attention (MLA) algorithm. These models differ in hidden dimensions, head dimensions, and number of heads.

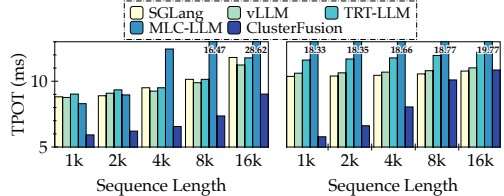
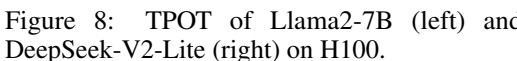
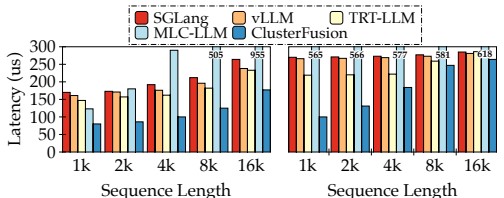

Figure 8: TPOT of Llama2-7B (left) and DeepSeek-V2-Lite (right) on H100.

Figure 9: Latency of core modules of Llama2-7B (left) and DeepSeek-V2-Lite (right) on H100.

## 4.1 End-to-End and Core Module Evaluation

We use time per output token (TPOT) as the metric for end-to-end evaluation. The results are presented in Fig. 17. For baselines, the results include both CUDA graph launch overhead and kernel execution latency. On average, ClusterFusion achieves 1.41×, 1.39×, 1.43×, and 2.03× speedups over SGLang, vLLM, TensorRT-LLM, and MLC-LLM across various sequence lengths with a cluster size of 4 on Llama2-7B. For DeepSeek-V2-Lite, ClusterFusion delivers average speedups of 1.34×, 1.37×, 1.51×, and 2.39× under the same conditions.

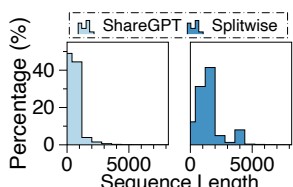

As shown in Fig. 18, for the core *QKV Projection*, *Attention*, and *Output Projection* modules, ClusterFusion achieves average speedups of 1.85×, 1.73×, 1.61×, and 3.19× compared to SGLang, vLLM, TensorRT-LLM, and MLC-LLM, respectively on Llama2-7B. Similarly, for DeepSeek-V2-Lite, ClusterFusion delivers average speedups of 1.66×, 1.64×, 1.35×, and 3.5×. For the DeepSeek MLA, which is specifically designed to better leverage GPU hardware, the optimization space for operator fusion is relatively limited. Nevertheless, as shown in Fig. 10, sequence lengths in real datasets are predominantly under 8k. In this range, ClusterFusion achieves significant performance improvements, demonstrating its effectiveness in real-world scenarios.

Figure 10: Sequence length distribution in ShareGPT [1] and Splitwise [41, 2] datasets.

We further evaluate the performance of the core modules in ClusterFusion under varying cluster sizes and numbers of *Attention* heads, with sequence lengths of 4K and 16K. The results are presented in Fig. 11. In our design, each *Attention* head is mapped to a cluster, so the cluster size determines the internal parallelism within each head. When the number of *Attention* heads is 32 and 64, a cluster size of 4 yields the best performance. However, when the number of heads increases to 128, a smaller cluster size of 2 becomes optimal. Conversely, cluster sizes of 8 and 16 lead to worse performance due to increased interconnect latency, bandwidth contention, and a reduced number of active SMs, which collectively limit overall core utilization, as illustrated in Fig. 5. Based on both theoretical analysis and empirical evidence, we conclude that the optimal cluster size varies across workloads. Therefore, cluster size should be tuned accordingly to maximize performance.

## 4.2 Speedup Analysis

We identify two primary factors contributing to the performance advantage of ClusterFusion over existing frameworks with CUDA Graph optimizations: minimized global memory transfer size and

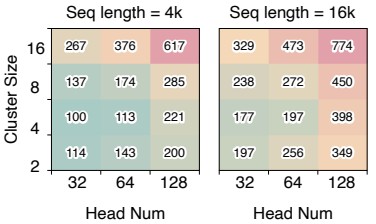

Figure 11: Latency of core module in ClusterFusion with varying settings.

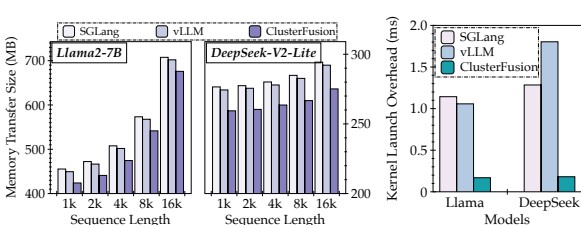

Figure 12: Comparison of global memory data transfer size (left) and GPU kernel launch overhead (right).

reduced GPU kernel launch overhead [63, 53]. To quantify these benefits, we leverage NVIDIA Nsight Systems [38] and Nsight Compute [37] to profile global memory transfer volume and GPU kernel launch overhead across different models and configurations. The results are illustrated in Fig. 19. The performance gain stems from the fact that `ClusterFusion` executes *QKV Projection*, *Attention*, and *Output Projection* entirely on-chip, significantly reducing intermediate memory traffic. Additionally, `ClusterFusion` reduces kernel launch overhead by nearly an order of magnitude in end-to-end scenarios, even when compared to baselines optimized with CUDA Graph.

## 4.3 Additional Analysis

Table 1: Latency comparison of on-chip ClusterReduce and ClusterGather with DSMEM versus off-chip implementations without DSMEM.

| Operation | Data Size ($KB$) | Off-chip ($\mu s$) | On-chip ($\mu s$) | Speedup |
|---|---|---|---|---|
| ClusterReduce | 32 | 8.03 | 6.77 | 1.18× |
| | 64 | 9.01 | 6.61 | 1.36× |
| | 128 | 14.95 | 7.42 | 2.01× |
| | 256 | 22.44 | 9.17 | 2.44× |
| ClusterGather | 32 | 6.26 | 3.90 | 1.60× |
| | 64 | 6.27 | 4.12 | 1.52× |
| | 128 | 6.31 | 4.39 | 1.44× |
| | 256 | 6.61 | 4.15 | 1.59× |

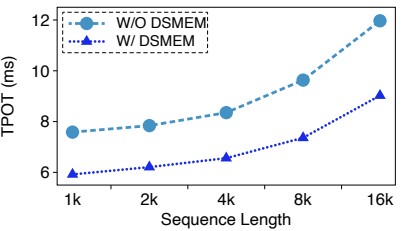

Figure 13: TPOT of `ClusterFusion` on Llama2-7B with and without DSMEM.

To demonstrate the importance of the on-chip interconnect leveraged in our dataflow design, we conduct a microbenchmark to evaluate the cluster-level collective communication primitives introduced in Sec. 3.1. As shown in Tbl. 1, the on-chip ClusterReduce and ClusterGather operations over DSMEM exhibit significantly lower latency across varying data transfer sizes compared to the off-chip implementations.

We further perform ablation studies to compare `ClusterFusion` with and without DSMEM enabled. The experimental results are presented in Fig. 13. Across different sequence lengths, disabling DSMEM increases the time per output token (TPOT) by up to 33%. These results highlight the effectiveness of DSMEM and the cluster-level collective primitives in enabling efficient on-chip reduction and aggregation, thereby improving end-to-end inference performance.

## 5 Discussion on Fusion Scope and Architectural Outlook

`ClusterFusion` builds on intra-cluster DSMEM communication, where each fused scope is bounded by a fixed cluster size (up to 16 thread blocks) [33]. This imposes constraints on fusion granularity and scheduling flexibility. Although most decoding operators in today's mainstream LLMs [51, 11], such as *Projection* and *Attention*, fit comfortably within this limit, future models with larger hidden dimensions or specialized operator variants may challenge this boundary. When fused operators exceed the cluster scope, the system must fall back to global memory communication, introducing additional latency and runtime fragmentation. This motivates reflection on hardware support for broader intra-chip collectives [20, 6, 27, 18].

Our findings suggest that enabling low-latency, topology-aware communication across a broader set of SMs would unlock more uniform and scalable fusion strategies. Such architectural support could extend structured coordination beyond current cluster boundaries. `ClusterFusion` highlights this co-design opportunity as a practical step toward supporting the growing complexity of architectures.

## 6 Conclusion

This paper presents `ClusterFusion`, an execution framework that schedules communication and computation jointly to expand operator fusion scope by composing operators during decoding stages such as *QKV Projection*, *Attention*, and *Output Projection* into a single fused kernels. By incorporating cluster-level collective communication primitives, `ClusterFusion` effectively reduces global memory traffic and kernel launch overhead, enabling efficient on-chip execution of key LLM decoding modules. Our comprehensive evaluation on NVIDIA H100 GPUs with representative

models Llama-2-7B and DeepSeek-V2-Lite demonstrates that `ClusterFusion` outperforms state-of-the-art LLM inference frameworks across different models and configurations.

## 7 Acknowledgement

This work was supported by the National Key R&D Program of China under Grant 2022YFB4501400, and the National Natural Science Foundation of China (NSFC) grant 62222210. This work was also supported by Shanghai Qi Zhi Institute Innovation Program SQZ202316. We sincerely thank Yilu Huang and Chiheng Jin for their assistance with kernel implementation and end-to-end integration. We would also like to thank the anonymous reviewers, as well as the Area Chairs and Program Chairs, for their constructive feedback and valuable comments that helped improve this work.

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

# A    Related Works

**Operator-Level Optimizations**    Numerous researchs have explored operator-level optimizaitons for LLM inference [60, 30, 28, 21, 61, 62]. FlashAttention [9, 8] fuses the entire Attention operation into a single memory-efficient kernel. Building on this, FlashDecoding [10] extends parallelism to the KV cache sequence dimension during decoding. FlashDecoding++ [19] proposes to determine the scaling factor based on statistics in advance and introduces FlatGEMM to represent the GEMM with a highly reduced dimension in decoding. FlashAttention-3 [44] demonstrated that warp specialization and new hardware features [33] such as asynchrony can have a significant impact on the Attention. FlashMLA [22] design an efficient decoding Attention kernel for DeepSeek MLA architecture inspired by FlashAttention-3. However, there optimizaitons adopt a block-isolated execution pattern. Due to the lack of structured communication across thread blocks, intermediate results are repeatedly written to and read from global memory, limiting opportunities for broader operator fusion and on-chip reuse. `ClusterFusion` explores a more general fusion space enabled by efficient cluster-scoped collective primitives.

**Algorithm-Level Optimizations**    Several studies focus on improving LLM inference efficiency through algorithmic changes [26, 29, 56, 31]. Techniques such as quantization and sparsification aim to reduce computational and memory overhead. Quantization [26, 29] compresses model weights and activations by converting high-bitwidth representations into lower-bitwidth ones, reducing arithmetic and memory costs. Pruning [56, 31] increases the proportion of zero elements in weights or activations, enabling hardware to skip redundant computations. These techniques are orthogonal to our work. `ClusterFusion` does not modify the model workload but can complement these approaches by optimizing the underlying dataflow and kernel execution through structure and reusable primitives.

**Systems on Inter-Core Connected Hardwares**    The on-chip inter-core interconnect has also been adopted in alternative hardware platforms such as Graphcore IPU [20] and Cerebras WSE [6]. Several prior works have explored leveraging this capabilities to optimize deep learning workloads. T10 [27] is an end-to-end deep learning compiler targeting inter-core connected intelligence processors, with an emphasis on fine-grained modeling of data movement between cores. WaferLLM [18], on the other hand, is the first system to propose a LLM parallelism solution tailored for wafer-scale accelerators. Moreover, these systems are closely tied to custom hardware architectures that offer full inter-core connectivity or integrates inter-core communication mechanisms into specific operators like GEMM and GEMV. As such, their design assumptions do not translate well to modern GPU architectures like NVIDIA Hopper which remains the dominant platform for LLM inference deployment. `ClusterFusion` introduces cluster-scoped communication primitives which abstract common aggregation and reduction patterns and can be flexibly reused across different workloads.

# B    Additional Dataflows

In this section, we first introduce the computation process of DeepSeek MLA and present our fused MLA dataflow, which leverages cluster-level collective communication primitives, as evaluated in the main paper. We then describe an alternative dataflow in which thread blocks partition the head dimension within the *Attention* module, in contrast to the dataflow in the main paper that partitions the KV cache along the token dimension, demonstrating the capability of our primitives in enabling different dataflows for the same computation. Finally, we evaluate these dataflow variants by analyzing their corresponding DSMEM traffic.

## B.1    DeepSeek MLA

DeepSeek introduces the Multi-Head Latent Attention (MLA) mechanism, which has been adopted in its model series [11, 12]. During inference, a weight absorption optimization[43] is applied in the decoding stage of MLA to reduce overall computational cost. The detailed computation processes of both the original MLA and the optimized version in DeepSeek-V2-Lite[11] are illustrated in Fig. 14. In the original MLA computation, the input hidden state first undergoes a *Q Projection* to generate the multi-head **Q**, and a *Down Projection* to obtain the compressed KV cache. The newly generated compressed KV is then concatenated with the cached KV and passed through an *Up Projection* to produce the multi-head **KV**, which is used in the MHA module. In the weight absorption version of

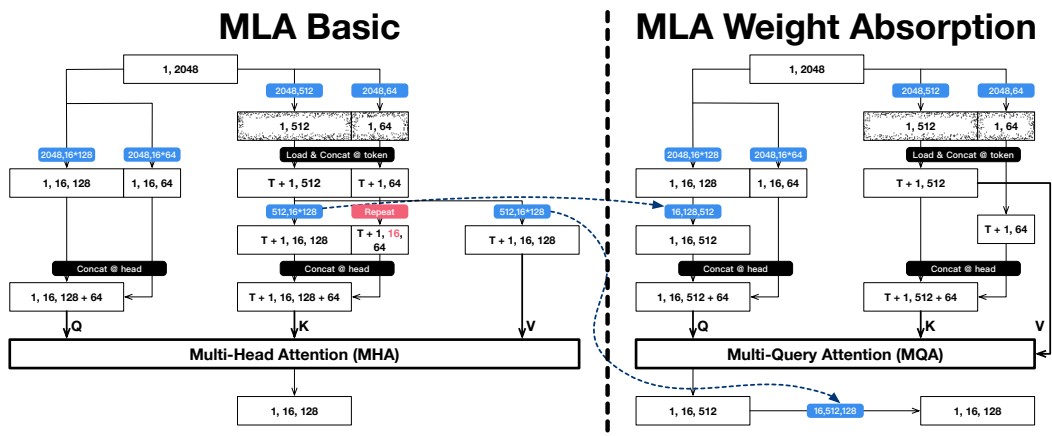

Figure 14: Overview of MLA computation: original (left) and with weight absorption optimization (right). The black stipple represents the newly generated latent presentation, which will be cached to calculate keys and values for *Attention* computation.

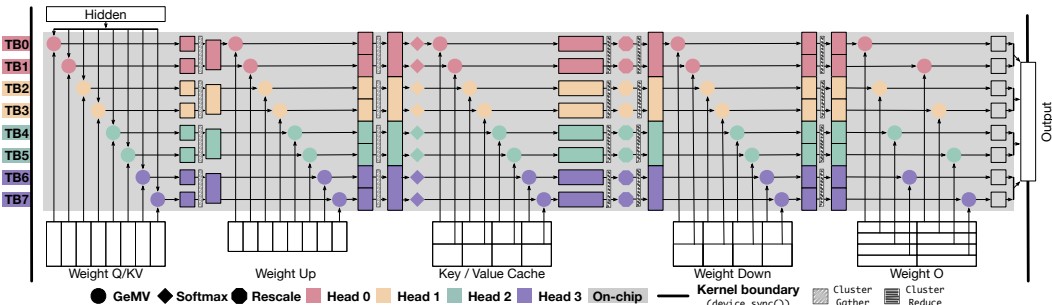

Figure 15: Cluster-centric fused MLA dataflow graph.

MLA, the process begins by computing $\mathbf{Q}$ and $\mathbf{K}$, where $\mathbf{V}$ is obtained by partially reusing $\mathbf{K}$:

$$Q = \text{Hidden} \times W_Q \times W_{\text{Up}}, \quad K = \text{Hidden} \times W_K, \quad V = K[: \texttt{kv\_lora\_rank}] \tag{3}$$

Next, the *Attention* module is computed as:

$$Z = \text{Concat}\left(\text{Softmax}\left(\frac{Q_1 K^\top}{\sqrt{d_k}}\right) V, \ldots, \text{Softmax}\left(\frac{Q_h K^\top}{\sqrt{d_k}}\right) V\right) \tag{4}$$

Finally, the result is passed through a *Down Projection* to produce the final *Attention* output:

$$\text{Output} = Z \times W_{\text{Down}} \tag{5}$$

The key difference between MLA and conventional MHA lies in the introduction of additional projection layers, specifically the *Up Projection* and *Down Projection*, which are designed to preserve mathematical consistency. Additionally, MLA employs the MQA mechanism [46], in which all $\mathbf{Q}$ heads share a single $\mathbf{KV}$ cache head. This approach increases computational intensity while reducing the memory access associated with the $\mathbf{KV}$ cache. Moreover, the head dimension in MLA corresponds to the value of `kv_lora_rank`, which is typically larger than that in other models. For example, in DeepSeek-V2-Lite, it is 512, whereas in Llama2-7B, the head dimension is 128.

According to our cluster-centric dataflow design principle, the fused MLA dataflow is illustrated in Fig. 15. This dataflow performs the entire MLA computation on-chip, eliminating any intermediate off-chip memory traffic. The detailed algorithm is presented in Alg. 4. This dataflow is parallelized across attention heads, with each head assigned to a cluster consisting of $N = 2^k$ thread blocks ($k \leq 4$). Each thread block within a cluster is assigned a rank $b \in [0, N-1]$. Within each cluster, thread blocks collaboratively partition the head dimension for the *QKV Projection*; the

---

**Algorithm 4** Fused MLA Dataflow - Thread Block View

---

**Require:** Input hidden states $\mathbf{H}_b \in \mathbb{R}^{1 \times D}$, *Q Projection* weight $\mathbf{W}_b^Q \in \mathbb{R}^{D \times h}$, *KV Projection* weight $\mathbf{W}_b^{KV} \in \mathbb{R}^{D \times l}$, *Up Projection* weight $\mathbf{W}_b^{Up} \in \mathbb{R}^{H \times l}$, *Down Projection* weight $\mathbf{W}_b^{Down} \in \mathbb{R}^{l \times H}$, *Output Projection* weight $\mathbf{W}_b^O \in \mathbb{R}^{H \times d}$, and KV cache $\mathbf{KV}_b^{\text{cache}} \in \mathbb{R}^{s \times L}$ in global memory.

1: Allocate shared memory buffers: $\mathbf{Q}_b, \mathbf{KV}_b \in \mathbb{R}^{1 \times l}$, and $S_{\text{sum}}, S_{\text{max}}$ (softmax statistics).
2: Compute segment results of *Q Projection*: $\mathbf{Q}_b \leftarrow \mathbf{H}_b \times \mathbf{W}_b^Q$.
3: Compute segment results of *KV Projection*: $\mathbf{KV}_b \leftarrow \mathbf{H}_b \times \mathbf{W}_b^{KV}$.
4: Obtain the complete QKV: $\mathbf{Q}_b \leftarrow \texttt{ClusterGather}(\mathbf{Q}_b), \mathbf{KV}_b \leftarrow \texttt{ClusterGather}(\mathbf{KV}_b)$.
5: Compute segment results of *Up Projection* by batch matmul: $\mathbf{Q}_b \leftarrow \mathbf{Q}_b \times \mathbf{W}_b^{Up}$.
6: Obtain the complete Q: $\mathbf{Q}_b \leftarrow \texttt{ClusterGather}(\mathbf{Q}_b)$.
7: Compute partial result of *Attention* similar to the FlashDecoding dataflow:

    Compute $\mathbf{S}_b \leftarrow \exp(\mathbf{Q}_b \times (\mathbf{KV}_b^{\text{cache}}, \mathbf{KV}_b)^T)$, obtain local $S_{\text{sum}}, S_{\text{max}}$.
    And store $S_{\text{max}}$ in register $Reg_{\text{max}}$.            ▷ softmax statistics.
    Compute $\mathbf{A}_b \leftarrow \mathbf{S}_b \times (\mathbf{KV}_b^{\text{cache}}, \mathbf{KV}_b)$    ▷ $\mathbf{A}_b$ reuse the shared memory space of $\mathbf{Q}_b$

8: Obtain the complete softmax statistics $S_{\text{sum}}$ and $S_{\text{max}}$:

$$S_{\text{sum}} \leftarrow \texttt{ClusterReduce}(S_{\text{sum}}, \text{sum}), \quad S_{\text{max}} \leftarrow \texttt{ClusterReduce}(S_{\text{max}}, \text{max})$$

9: Rescale *Attention* output according to online softmax:

$$\mathbf{A}_b \leftarrow \frac{\mathbf{A}_b \cdot \exp(Reg_{\text{max}} - S_{\text{max}})}{S_{\text{sum}}}$$

10: Obtain the complete *Attention* output: $\mathbf{A}_b \leftarrow \texttt{ClusterReduce}(\mathbf{A}_b, \text{sum})$.
11: Compute partial results of *Down Projection* by batch matmul: $\mathbf{A}_b \leftarrow \mathbf{A}_b \times \mathbf{W}_b^{Down}$.
12: Obtain the complete *Down Projection* output: $\mathbf{A}_b \leftarrow \texttt{ClusterReduce}(\mathbf{A}_b, \text{sum})$.
13: Compute the result of the *Output Projection* and write it to global memory using $\texttt{atomicAdd}$:

$$\mathbf{O}_b \leftarrow \mathbf{A}_b \times \mathbf{W}_b^O$$

14: **return** $\mathbf{O}_b$.

---

`kv_lora_rank` dimension for the *Up Projection* and *Down Projection* modules; the token dimension of the KV cache for the *Attention* module; and the output dimension for the *Output Projection*. And each thread block processes the full input hidden states and computes its corresponding output tile $O_b$ after the *Output Projection*. In this algorithm, $B$ denotes the batch size, $D$ the input hidden dimension, and $H$ the total head dimension. The variables $h$, $l$, $s$, and $d$ refer to the partitioned sizes of the head dimension, `kv_lora_rank`, sequence length, and output dimension per thread block, respectively. For simplicity, we omit the `rope_dim` shown in Fig. 14.

We also estimate the total DSMEM traffic incurred by the dataflow used in Alg. 4, which involves three `ClusterGather` operations and three `ClusterReduce` operations. The DSMEM traffic for the `ClusterGather` operations is given by:

$$Traffic_{Gather}(h, N) + 2 \times Traffic_{Gather}(l, N)$$

and the traffic for the `ClusterReduce` operations is:

$$Traffic_{Reduce}(l, N) + Traffic_{Reduce}(H, N)$$

We omit the traffic introduced by the softmax statistics, as it involves only two floating-point values and is negligible compared to the tensor-level data movement.

## B.2 SplitHead Dataflow

In line with our cluster-centric dataflow design principles and by leveraging cluster-level communication primitives, we implement the fused *QKV Projection*, *Attention*, and *Output Projection* dataflow described in the main paper, where intermediate data is stored in on-chip shared memory. In addition, we design an alternative dataflow called SplitHead dataflow that stores the intermediate data in faster

**Algorithm 5** SplieHead Dataflow - Thread Block View

---

**Require:** Input hidden states $\mathbf{H}_b \in \mathbb{R}^{B \times D}$, *QKV Projection* weight $\mathbf{W}_b^{QKV} \in \mathbb{R}^{D \times 3h}$, *Output Projection* weight $\mathbf{W}_b^O \in \mathbb{R}^{h \times D}$, and KV cache $\mathbf{K}_b^{\text{cache}}, \mathbf{V}_b^{\text{cache}} \in \mathbb{R}^{S \times h}$ in global memory.

1: Allocate register memory buffers: $\mathbf{Q}_b, \mathbf{K}_b, \mathbf{V}_b \in \mathbb{R}^{B \times h}$, shared memory buffer: $\mathbf{S}_b \in \mathbb{R}^{S \times B}$.
2: Compute segment results of *QKV Projection*: $\mathbf{Q}_b, \mathbf{K}_b, \mathbf{V}_b \leftarrow \mathbf{H}_b \times \mathbf{W}_b^{QKV}$.
3: Compute segment result of *Attention*:
      Compute $\mathbf{S}_b \leftarrow \mathbf{Q}_b \times (\mathbf{K}_b^{\text{cache}}, \mathbf{K}_b)^T$.
      Obtain the complete result of $\mathbf{Q} \times \mathbf{K}^T$: $\mathbf{S}_b \leftarrow \texttt{ClusterReduce}(\mathbf{S}_b, \text{sum})$.
      Compute $Softmax$: $\mathbf{S}_b \leftarrow softmax(\mathbf{S}_b)$.
      Compute *Attention* output: $\mathbf{A}_b \leftarrow \mathbf{S}_b \times (\mathbf{V}_b^{\text{cache}}, \mathbf{V}_b)$
4: Compute the partial one head result of the *Output Projection*: $\mathbf{O}_b \leftarrow \mathbf{A}_b \times \mathbf{W}_b^O$.
5: Obtain the complete one head result of the *Output Projection*:

$$\mathbf{O} \leftarrow \texttt{ClusterReduce}(\mathbf{O}_b, sum)$$

6: Write the complete result of the *Output Projection* to global memory using `atomicAdd`:

$$\mathbf{O} \leftarrow \texttt{atomicAdd}(\mathbf{O})$$

7: **return O**.

---

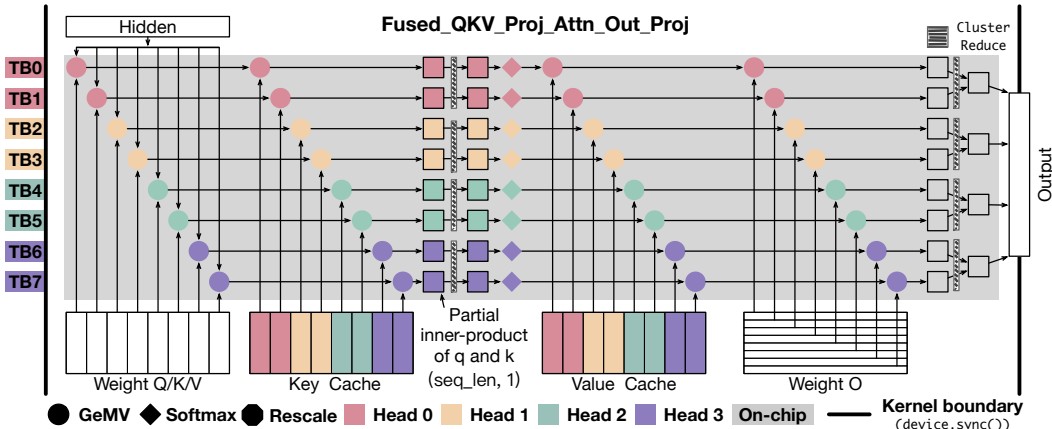

Figure 16: Cluster-centric fused *QKV Projection*, *Attention* and *Output Projection* dataflow graph.

on-chip register memory. As illustrated in Alg. 5, this dataflow is parallel in the number of heads. Each head corresponds to a cluster of $N = 2^k$ thread blocks ($k \le 4$), where each block is assigned a rank $b \in [0, N-1]$. Within each cluster, thread blocks just partition the head dimension in *QKV Projection*, *Attention*, and *Output Projection*. For the whole dataflow, each thread block processes the entire input hidden states and computes the corresponding partial output $O_b$ after the *Output Projection*. In this algorithm, $B$ denotes the batch size, $D$ the input hidden dimension, $H$ the total head dimension, $S$ the KV cache sequence length, and $h, d$ represent the partitioned sizes of the head dimension and the output dimension, per thread block, respectively. In this design, we need to reduce the result of $\mathbf{Q} \times \mathbf{K}^T$ which has a shape of $Sequence\ Length \times Batch\ Size$. Each thread block holds the segment *Attention* output and computes the partial *Output Projection*, which must then be reduced and written to global memory using `atomicAdd`.

As shown in Fig. 16, this dataflow enables intermediate results such as $\mathbf{Q}_b$, $\mathbf{K}_b$, and $\mathbf{V}_b$ to be stored in faster register memory, instead of shared memory. However, the total DSMEM traffic incurred by this dataflow is

$$Traffic_{Total} = Traffic_{Reduce}(S, N) + Traffic_{Reduce}(D, N)$$

which is higher than the dataflow that partitions the KV cache along the token dimension in the *Attention* module, as proposed in the main paper and Sec. B.1. The total DSMEM traffic of the

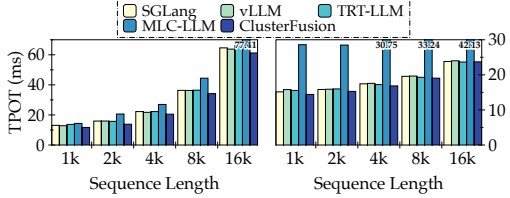
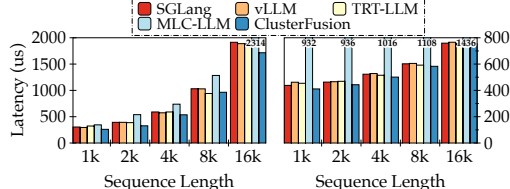

Figure 17: TPOT of Llama2-7B (left) and DeepSeek-V2-Lite (right) on H100.

Figure 18: Latency of core modules of Llama2-7B (left) and DeepSeek-V2-Lite (right) on H100.

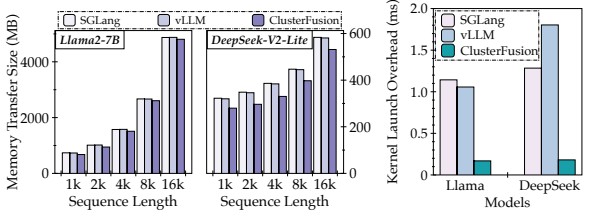
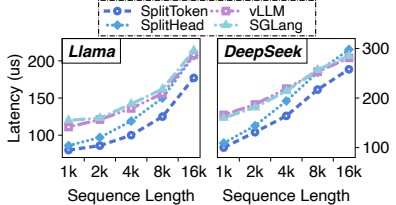

Figure 19: Comparison of global memory data transfer size (left) and GPU kernel launch overhead (right).

Figure 20: Latency comparision of SplitToken and SplitHead dataflows.

SplitToken dataflow is

$$Traffic_{Total} = Traffic_{Reduce}(H, N) + Traffic_{Gather}(3h, N)$$

which is significantly lower, as it mainly depends on the head dimension $H$ or the `kv_lora_rank` $l$ in the fused MLA dataflow, both of which are much smaller than the sequence length $S$ that dominates the DSMEM traffic in the SplitHead dataflow. The increased communication overhead outweighs the benefits of using register memory. As demonstrated by our experimental results presented later, the SplitHead dataflow yields higher latency.

## C  Additional Experiments

### C.1  Multi-Batch Evaluation and Speedup Analysis

We conduct additional experiments using a batch size of 16, while keeping other experimental settings consistent with those described in the main paper. The TPOT and latency results for core modules are shown in Fig. 17 and Fig. 18, respectively. On average, `ClusterFusion` achieves speedups of 1.11×, 1.09×, 1.12×, and 1.32× over SGLang, vLLM, TensorRT-LLM, and MLC-LLM, respectively, across various sequence lengths on Llama2-7B with a cluster size of 4. For DeepSeek-V2-Lite, `ClusterFusion` delivers average speedups of 1.15×, 1.14×, 1.07×, and 1.84× under the same conditions. In terms of the core modules, including fused *QKV Projection*, *Attention*, and *Output Projection*, `ClusterFusion` achieves average speedups of 1.14×, 1.12×, 1.2×, and 1.41× over SGLang, vLLM, TensorRT-LLM, and MLC-LLM, respectively, on Llama2-7B. Similarly, for DeepSeek-V2-Lite, the corresponding speedups are 1.19×, 1.18×, 1.14×, and 2.04×.

We identify two primary factors contributing to the performance advantage of `ClusterFusion` over existing frameworks with CUDA Graph optimizations: reduced global memory transfer volume and significantly lower GPU kernel launch overhead [63, 53]. To quantify these benefits, we use NVIDIA Nsight Systems [38] and Nsight Compute [37] to profile memory traffic and kernel launch overhead under a batch size of 16. The results are presented in Fig. 19. The observed performance gains are primarily attributed to `ClusterFusion` executing *QKV Projection*, *Attention*, and *Output Projection* entirely on-chip, thereby mi nimizing intermediate memory traffic. Furthermore, `ClusterFusion` reduces kernel launch overhead by nearly an order of magnitude in end-to-end scenarios, even compared to baselines already optimized with CUDA Graph. However, the reduction in global memory traffic has limited impact in the multi-batch scenario, as the KV cache and model weights still dominate memory usage, while the intermediate memory footprint remains small. Moreover, the overall computation intensity increases significantly with larger batch sizes, leading to a reduced speedup compared to the single-batch results presented in the main paper.

## C.2 SplitHead Dataflow Evaluation

We also implement the SplitHead dataflow described in Sec. B.2 and present the performance comparison between the SplitToken, SplitHead dataflows and two representative baselines in Fig. 20. When the sequence length is short, the latency difference is minimal because intermediate data can be stored in register memory, which improves efficiency compared to the SplitToken, and the gap in DSMEM traffic compared to the SplitToken dataflow remains small. However, as the sequence length increases, the DSMEM traffic of the SplitHead dataflow grows significantly larger than that of SplitToken, resulting in increased latency. From the perspective of operator fusion efficiency, the SplitHead dataflow enables fusion with intermediate data stored in high-speed register memory, as long as the data size is small enough to fit entirely within the registers. However, despite this benefit, the SplitHead dataflow incurs significantly higher DSMEM traffic, especially as the sequence length increases. This increased traffic becomes a major bottleneck and leads to worse overall performance. Therefore, our cluster-centric dataflow design takes into account not only fusion efficiency but also the memory traffic of different dataflows, aiming to achieve better overall performance.

