# OpenReview forum: "ClusterFusion: Expanding Operator Fusion Scope for LLM Inference via Cluster-Level Collective Primitive"
_NeurIPS.cc/2025/Conference — NeurIPS 2025 poster_

### Official Review · Reviewer_1tSz · 2025-06-15

**Clarity:** 3
**Significance:** 3
**Originality:** 3
**Rating:** 5
**Confidence:** 5

**Summary:**

This paper applied the idea of "collective communication" or tensor parallellism/sharding between multiple devices to the sharding/on-chip collectives between threading blocks/Streaming Processors within a single device, and demonstrates it enables kernel fusion of the entire self-Attention module and performance gain over SOTA inference kernels/engines.

**Questions:**

1. It is a bit confusing to use the term "Cluster" as it typically refers to distributed multi-device/node system. While the core ideas of collectives are the same with a cluster of devices and a cluster of threadblocks, it may be a bit misleading for readers. Similarly the authors called the two primitives Cluster-Reduce/Gather, aren't they just Block/SM level all-reduce/gather?

2. The collectives are implemented via tree pattern, have you considered ring pattern?

3. How do you define the bandwidth between a cluster of SMs? is it defined by peer-to-peer bw if so I am not sure why it is affected by cluster size or is it deifned by bipartite or all-reduce bw etc?

4. Why would the active SMs descrease with size of clusters?

5. Line 256 mentioned the kernel launch overhead is reduced by an order, however based on Figure 7, the kernel is a fusion of 4 kernels of Figure 3, how is it able to reduce kernel launch overhead by an order (or 10 times)?

**Ethical Concerns:**

["NO or VERY MINOR ethics concerns only"]

**Final Justification:**

The rebuttal addressed most of my questions with detailed explanations.

**Limitations:**

* While included in appendix, I think the authors should discuss the efficacy of the method on higher batchsize > 1 in the main body. e.g., why does higher batch size see a much less performance gain over baselines when no weight is involved (unlike MLPs) so off-chip traffic is purely proportional to batchsize

* The onchip collectives rely on creating buffers with onchip shared-memory, what if the model/kernel is too large so it runs out of the pretious onchip memory?

**Quality:**

3

**Strengths And Weaknesses:**

Strength:
* The motivaiton is intuitive and novel on GPU programming: modern LLM decoding on GPU is bottlenecked by off-chip memory bandwidth and kernel launch/synchronization overhead. And the onchip collective communication between SMs/Thread-blocks and reduce off-chip traffic and promote kernel fusions

* The author abstract and implemented two effective collective communications among a thread block cluster: All-Reduce and All-Gather, which serves the fundamental to perform tensor/operator sharding onchip

* The authors demonstrated significant performance gain e.g., 50% over SOTA inference engine/kernels with BS=1

Weakness:
* While relative new among GPU, onchip datalfow/collectives have been widely explored by dataflow architecture accelerators like cited GraphCore/Cerebras etc, the associated aggressive kernel fusion has also been studied by https://arxiv.org/pdf/2410.23668 which extends kernel fusion to the entire transformer model. Recently there is also research about mega kernel fusion on GPU: https://hazyresearch.stanford.edu/blog/2025-05-27-no-bubbles, which should be included in discussion

- The authors didn't discuss enough limitations of this approach, detailed below

---

> ### Author Rebuttal · Authors · 2025-07-30
>
> Thank you for your thoughtful feedback and suggestions on our paper. We would like to provide more clarification.
>
> To answer your concerns:
> 1. **About the term "Cluster".** Sorry about the confusion, we will use the term "SM Cluster" to avoid this confusion. Indeed, our primitives operate at the block/SM level. We chose the term “Cluster” to align with NVIDIA’s terminology on Hopper GPUs[1,2], where a cluster refers to a group of thread blocks with shared memory connectivity.
> 2. **Different communication pattern.** Thanks for your point. We have carefully considered both the tree pattern and the ring pattern (ScatterReduce + AllGather), and we chose the tree pattern because it requires fewer communication rounds, which leads to lower latency when the data size is relatively small.
> We compared the two patterns both theoretically and empirically under cluster size = $N$ and data size = $K$. In theory, the ring pattern requires $(N-1)$ rounds for ScatterReduce and another $(N-1)$ rounds for AllGather, totaling $2(N - 1)$ communication rounds, with each round transferring $\frac{K}{N}$ data per node. In contrast, the tree pattern only needs $\log_2 N$ rounds, with each round transferring $K$ data per node, as described in our paper.
> The following table summarizes the theoretical communication characteristics for both patterns:
>
> | Pattern     | Comm Rounds     | Data per Round | Total Data per Node |
> |-------------|------------------|----------------|----------------------|
> | Ring        | 2(N - 1)         | K / N          | 2(N - 1) * (K / N)   |
> | Tree        | log₂(N)          | K              | K * log₂(N)          |
>
> For example, when $N=2$ and $N=4$:
>
> | Pattern \ N | Rounds | Data/Round | Total Data |
> |-------------|--------|------------|-------------|
> | Ring (N=2)  | 2      | K/2        | K           |
> | Tree (N=2)  | 1      | K          | K           |
> | Ring (N=4)  | 6      | K/4        | 1.5K        |
> | Tree (N=4)  | 2      | K          | 2K          |
>
> Although the tree pattern transfers more total data, its fewer communication rounds are more favorable for latency because the data size $K$ is small (64–128 bytes) in the ClusterFusion dataflow and DSMEM bandwidth is not fully utilized. This is also reflected in our empirical results, where we report the kernel latency(us) of the core ClusterFusion kernel:
> | Pattern\Cluster Size | 2 | 4 |
> |------|------|------|
> | Ring | 135 | 127 |
> | Tree | 132   | 125 |
>
> We plan to include a more detailed discussion about different communication pattern in the final version.
> 3. **Definition of the bandwidth between SMs within cluster.** The bandwidth shown in Section 2.3 and Figure 5 refers to peer-to-peer bandwidth. As cluster size increases, peer-to-peer bandwidth decreases. Based on the patent and paper referenced in that section, the interconnect topology for DSMEM in Hopper GPUs is likely a crossbar or crossbar-like architecture, which allows any SM to communicate with any other within a cluster. While a crossbar provides full connectivity, it does not provide dedicated bandwidth for all possible communication pairs simultaneously. Instead, the internal paths of the crossbar must be shared among all active transfers, and contention arises as more thread blocks in a cluster attempt concurrent communication.
>
> 4. **The reason for the decreased active SMs.** To the best knowledge of us based on public architectural information[1,2,3], the number of active SMs may decrease with larger cluster sizes due to hardware-level SM allocation constraints. Whenever the cluster size exceeds 2, the hardware grouping may lead to SM underutilization due to yield-related SM inactivation and cluster alignment issues. On H100 SXM GPUs, the 132 SMs are not evenly distributed: three GPCs[2] have 18 SMs, four have 16, and the last one contains 8 + 6 SMs, where the last 6 are special. For example, when using Cluster Size = 4, this grouping cannot fully utilize all SMs: in the first three GPCs, 2 SMs per GPC are left out (3 × 2 = 6), and in the last GPC, all 6 special SMs are likely excluded, totaling 12 unused SMs. This leaves only 120 active SMs which aligns with our observed drop in active SMs during profiling.
> 5. **About kernel launch overhead in Figure 7.** Thanks for pointing this out. Actually, Figure 7 shows the total kernel launch overhead of the end-to-end decoding. Our fused kernel reduces the overall kernel launch overhead across all layers. And our kernel-level launch overhead is indeed one order of magnitude lower than the CUDA Graph launch overhead used in existing frameworks. For example, in the Llama2-7B model, the end-to-end kernel launch overhead drops from 1.057 ms in the baseline to 0.169 ms with ClusterFusion.
> 6. **The performance of multi-batch.** For batch sizes >1, we observe an average 1.2× end-to-end speedup in the Appendix provided in the supplementary material. As batch size grows, the workload becomes more compute-intensive and less memory-bound, reducing the optimization space for memory traffic. Although the off-chip memory traffic increases in higher batch size, the proportion of activation memory traffic and sync decreases due to the significantly increased compute intensity. Nevertheless, ClusterFusion still achieves meaningful gains, demonstrating its robustness across diverse inference scenarios.
> 7. **The reliance on shared memory buffer.** In our current implementation, the algorithm relies on either reusing existing shared memory tiles (e.g., weight tiles) or allocating a small additional buffer. This is necessary because our reduction is single-phase: the data being transmitted and the data being reduced in each iteration reside in the same memory region, requiring intermediate buffering to avoid conflicts. However, when shared memory becomes insufficient, we can switch to buffer-free variants such as ScatterReduce. In this approach, each thread block writes its data to non-overlapping destinations, allowing communication and reduction to proceed simultaneously without intermediate buffering.
> 8. **Comparison with other concurrent work.** Thank you for pointing out two recent, concurrent work: Kernel Looping[4] and MegaKernel[5](publicly available after our submission). Kernel Looping is designed for a dataflow architecture accelerator—specifically the SambaNova SN40L Reconfigurable Dataflow Unit (RDU). It leverages the repetitive nature of decoder layers in LLM to fuse different layers, reducing multiple kernel launches to a single launch. This approach serves a role similar to CUDA Graphs in the GPU. Our method also significantly reduces kernel launch overhead on GPUs, and we empirically outperform frameworks that already leverage CUDA Graphs. The motivation of MegaKernel is similar to ours in that both aim to reduce memory bubble. Their approach is orthogonal to our work. They focus on fine-grained shared memory reuse and overlapping activation storage with weight loading. Despite this, MegaKernel still relies on global memory for intermediate result synchronization, whereas our work introduces cluster-level primitives to enable fast, on-chip synchronization. We will incorporate a detailed discussion of these recent works in the related work of the final paper.
>
> [1] NVIDIA CUDA C++ Programming Guide.
>
> [2] NVIDIA H100 Tensor Core GPU Architecture.
>
> [3] Benchmarking Thread Block Cluster, HPEC 2024.
>
> [4] Kernel Looping: Eliminating Synchronization Boundaries for Peak Inference Performance, arxiv.
>
> [5] Look Ma, No Bubbles! Designing a Low-Latency Megakernel for Llama-1B, Hazy Research.

---

> > ### Comment · Reviewer_1tSz · 2025-08-07
> >
> > Thanks for the detailed explanations. Regarding the kernel launch again, my understand is the kernel fusion is within a single Transformer layer not all the decoding layers so it is a fusion of 4 kernels that have global synchronizaiton. So I am still not clear where the 10 times reduction comes from. Regardless I am inclined to keep my score.

---

> ### Author Response · Authors · 2025-08-08
>
> Thanks for your further feedback, and we appreciate the opportunity to clarify the kernel launch details.
> 1. **Regarding the exact number of kernels.** Actually, the four modules shown in Figure 7 of our paper do not correspond to exactly four kernels in the baseline. Although existing SOTA frameworks already leverage torch.compile optimizations, there remain some fragmented kernels, as detailed in the tables below:
>
> For Llama:
> | Kernel Name     | Kernel Count | Kernel Description |
> |--------------|--------------|--------------|
> | torch.compile generated gemm | 2 | QKV Proj and Output Proj|
> | element-wise kernel| 4 |For tensor split, reshape, index adjustments and updates |
> | flashinfer::decode_attention | 1 |Attention (FlashDecoding dataflow)  |
> | flashinfer::merge_states | 1 |Attention output reduce|
>
> For DeepSeek:
> | Kernel Name                         | Kernel Count | Kernel Description                      |
> |--------------|--------------|--------------|
> | torch.compile generated gemm       | 4            | QKV Proj, Q Up Proj, Output Down Proj and Output Proj |
> | element-wise kernel                | 7            | For tensor split, reshape, index adjustments and updates |
> | flashinfer::decode_attention       | 1            | Attention (FlashDecoding dataflow)                            |
> | flashinfer::merge_states           | 1            | Attention output reduce               |
>
> This detailed kernel count information is also available in our supplementary materials. We will consider clarifying this in the experimental section.
>
> 2. **Regarding the precise speedup.** Thank you for pointing this out. Actually, our kernel launch latency reduction from 1.057ms to 0.169ms corresponds to about a 6.25× improvement, which is not a full order of magnitude. We originally used "nearly one order of magnitude lower" to emphasize the substantial decrease, but we apologize for the inaccurate phrasing that could be interpreted as a 10× improvement. We will revise this to a more precise description. And regarding Figure 12, the exact latency and corresponding speedup are as follows:
>
> For Llama:
> | Framework     | Kernel launch overhead (ms) |
> |--------------|-------------------------------|
> | SGLang       | 1.143 (6.76x)                         |
> | vLLM         | 1.057 (6.25x)                        |
> | ClusterFusion| 0.169                         |
>
> For DeepSeek:
> | Framework     | Kernel launch overhead (ms) |
> |--------------|-------------------------------|
> | SGLang       | 1.284 (7.09x)                        |
> | vLLM         | 1.803 (9.96x)                        |
> | ClusterFusion| 0.181                         |

---

> > ### Comment · Reviewer_1tSz · 2025-08-08
> >
> > Okay, thanks for the detailed explanation and given the clarity I decided to raise my score.

---

> > > ### Author Response · Authors · 2025-08-08
> > >
> > > We are pleased to hear that the detailed explanation addresses your concerns and thanks for appreciating our work and raising your score.

---

### Official Review · Reviewer_KUk8 · 2025-06-20

**Clarity:** 3
**Significance:** 2
**Originality:** 3
**Rating:** 5
**Confidence:** 4

**Summary:**

On NVIDIA Hopper GPUs, distributed shared memory is introduced to allow two thread blocks within the same thread block cluster to access each other’s shared memory. This feature provides a more efficient method for inter-block communication compared to using global memory. This paper proposes two cluster-level collective primitives that leverage this low-level feature. Based on the proposed primitives, the authors also implement a fused attention mechanism (including QKV projections, self-attention, and output projection) in a single kernel. Experiments are conducted to demonstrate the effectiveness of the proposed primitives and the fused kernel.

**Questions:**

Thank you to the authors for submitting to NeurIPS 2025. The paper is well written and easy to follow. Overall, I am positive about the submission. I have two concerns that I hope can help improve the paper.

1. **The proposed primitives themselves do not present much novelty. Reduce and gather are standard collective primitives.**
   While reduce and gather are standard primitives, building them using low-level PTX instructions may be a novel aspect of this work. Therefore, I recommend clarifying which low-level instructions are used (e.g., `ld.shared::cluster.u32`), possibly in a table, and explaining why they are difficult to use directly.

2. **More evidence or experiments are needed to justify the necessity of the proposed primitives for the fusion.**
   Figure 7 illustrates the fused attention operation using the proposed collective primitives. However, I did not see a fundamental limitation preventing similar kernel without using cluster-level primitives. For example, could the same thread block handle the tasks of both TB0 and TB1 in Figure 7 by reducing the tile size? If so, additional experiments comparing the block cluster approach with a single thread block would be helpful. If not, it would be beneficial to explain why, e.g., due to insufficient shared memory, with concrete numbers.

**Ethical Concerns:**

["NO or VERY MINOR ethics concerns only"]

**Final Justification:**

Thanks the authors for the detailed response which addressed majority of my concerns, increasing the score from 4 to 5.

**Quality:**

3

**Strengths And Weaknesses:**

**Strengths**

1. The proposed techniques target important workloads (i.e., efficient LLM inference) on relevant hardware (i.e., recent NVIDIA GPUs).
2. The results are promising.
3. The paper is well written and easy to follow.

**Weaknesses**

1. The proposed primitives themselves do not present much novelty. Reduce and gather are standard collective primitives.
2. More evidence or experiments are needed to justify the necessity of using the proposed primitives for the fusion.

---

> ### Author Rebuttal · Authors · 2025-07-30
>
> Thank you for your thoughtful feedback and suggestions on our paper. We would like to provide more clarification.
>
> To answer your concerns:
> 1. **About novelty and the low-level PTX.** While reduce and gather are common primitives in distributed system, they are rarely considered in chip-scale with interconnect. This enable new optimization scope and opportunities, and our novelty mainly lies in the exploration of on-chip dataflow via interconnect. On Hopper GPUs, blocks within a cluster are fully connected via DSMEM, as illustrated in Figure 4 and Figure 6. And as shown in Section 2.3 and Figure 5, DSMEM performance is non-trivial and not automatically efficient—naïve use often underperforms. Our primitives apply structured, tree-pattern, and collective scheduling to fully exploit DSMEM, enabling practical performance gains in real LLM workloads. And we have provided our codes in the supplementary material. We use the following set of PTX instructions in our implementation of ClusterReduce and ClusterGather:
>
> | High-level Function | PTX Instruction | Description |
> |---------------|---------------|---------------|
> | Init DSMEM barrier                  | `mbarrier.init.shared::cta.b64`                                | Initialize inter-block synchronization barrier within a CTA                |
> | Notify barrier with transfer data size    | `mbarrier.arrive.expect_tx.shared::cta.b64`                    | Notify barrier with the expected data size of Reduce/Gather           |
> | Map shared memory across blocks     | `mapa.shared::cluster.u32`                                     | Map the shared memory address from one thread block to another             |
> | DSMEM copy | `cp.async.bulk.shared::cluster.shared::cta.mbarrier::complete_tx::bytes` | Copy data across shared memory of blocks with barrier synchronization      |
> | Wait for DSMEM transfer completion | `mbarrier.try_wait.parity.shared::cta.b64`                     | Used by the consumer thread block to detect whether the producer has completed the DSMEM write            |
>
>
> All of these instructions are necessary for implementing our ClusterReduce and ClusterGather primitives. While these instructions are documented in PTX ISA references [1,2], they are low-level, semantically fragmented, and designed only for fine-grained behaviors (e.g., address mapping or signaling). Their correct composition into usable patterns is non-trivial. Many of them take complicated parameters (e.g., barrier state tokens, data sizes, memory space specifiers) and require strict ordering to avoid deadlocks or incorrect synchronization. Our contribution is to encapsulate these instructions into robust, reusable cluster-level primitives. This abstraction significantly reduces development cost and makes it practical to integrate DSMEM communication into LLM inference frameworks.
>
> 2. **The necessity of the proposed primitives for fusion.** Thanks for your insightful suggestion. Our cluster-level primitives are necessary to scale fusion across realistic LLM workloads with high parallelism. Indeed, it is possible to let a single thread block handle the tasks of both TB0 and TB1 in Figure 7 (equivalent to setting cluster size = 1), which eliminates the need for inter-block communication. We have implemented and evaluated this variant. The evaluation result of ClusterFusion core kernel latency(us) under different cluster size are:
>
> | Seqlen\Cluster Size | 4 | 2 | 1 (w/o Cluster-level Primitives) |
> |------|------|------|------|
> | 4k | 108 | 121 | **212** |
> | 8k | 125   | 132  | **224** |
>
> As above results show, using only one thread block significantly degrades performance. This is because a single block often lacks sufficient parallelism. In practice, we need to split the workload across more thread blocks (e.g., along the head or token dimension to fully utilize the hardware). However, such splits introduce inter-block data dependencies that require reduction or aggregation. This is where our on-chip ClusterReduce and ClusterGather primitives become critical: they enable efficient intra-cluster communication without falling back to global memory.
>
>
> [1] Parallel Thread Execution ISA Version 8.8, NVIDIA.
>
> [2] NVIDIA H100 Tensor Core GPU Architecture.

---

> > ### Comment · Reviewer_KUk8 · 2025-08-02
> >
> > Thanks for the table of how to use low-level PTX instructions to construct your primitives, as well as the numbers when not using the thread block cluster and distributed shared memory.

---

> ### Comment · Reviewer_KUk8 · 2025-08-05
>
> Raising my score from 4 to 5 since the response addressed majority of my concerns.

---

> > ### Author Response · Authors · 2025-08-06
> >
> > We are pleased to hear that the rebuttal addresses your concerns and thanks for appreciating our work and raising your score.

---

### Official Review · Reviewer_NWDf · 2025-07-02

**Clarity:** 3
**Significance:** 3
**Originality:** 3
**Rating:** 4
**Confidence:** 4

**Summary:**

This paper proposes ClusterFusion, a new system to make large language models (LLMs) run faster during inference (when generating tokens) by fusing multiple steps—like QKV projection, attention, and output projection—into a single, optimized computation block.

The key problem is that current LLM systems run these steps separately and move data to and from off-chip memory, which slows things down. Modern GPUs like NVIDIA Hopper offer on-chip memory and fast interconnects within thread block clusters, but these are hard to use efficiently due to low-level programming.

To solve this, the authors design two new primitives:
- ClusterReduce: for reducing values (e.g., sum or max) across thread blocks.
- ClusterGather: for sharing data between thread blocks.
These allow on-chip communication and help fuse multiple operators together in a more coordinated way. The authors implement these ideas into an inference framework called ClusterFusion, which they test on Llama2-7B and DeepSeek-V2-Lite using an NVIDIA H100 GPU. Compared to popular systems like vLLM, SGLang, and TensorRT-LLM, ClusterFusion achieves an average speedup of ~1.6× in decoding time, especially on long sequences.

**Questions:**

Questions & Suggestions for the Authors

1. How portable is ClusterFusion beyond NVIDIA Hopper?
Your method relies on Hopper-specific features, such as DSMEM and thread block clusters. Can your approach be adapted to other GPUs, e.g., A100?
2. What does integration with real-world LLM systems look like?
You claim speedups over SOTA frameworks like vLLM and TensorRT-LLM—but how would someone actually adopt ClusterFusion in production? Do you require kernel-level rewrites, custom graph compilers, or modified CUDA runtimes?
3. Why is fusion limited to three modules (QKV, Attention, Output)?
You call it “ClusterFusion,” but the fusion scope still ends after Output Projection. Why not include FFN, RMSNorm, or RotaryEmbeddings? Are there hardware barriers or software trade-offs you can explain more clearly? This would clarify how general and extensible the method really is.
4. Do you expect ClusterFusion to scale to batch sizes >1 or mixed-length prompts?
All your evaluations use batch size = 1. That’s common in interactive inference, but real serving systems also deal with batching and variable-length requests. Does your method still deliver speedup in those conditions? If not, could you clarify when it's useful vs. not useful?
5. How do you recommend users tune cluster size and layout in practice?
You provide detailed analysis of cluster size trade-offs, but no guidance on how a practitioner should make these decisions automatically. Could you expose this as a tunable parameter? Or integrate it into a profiling-based auto-tuner?

**Ethical Concerns:**

["NO or VERY MINOR ethics concerns only"]

**Limitations:**

1. Highly hardware-specific design
The entire framework is tightly coupled to NVIDIA Hopper GPUs. There is no discussion about compatibility with previous-generation GPUs (e.g., A100) or non-NVIDIA hardware (e.g., AMD, Intel).
The method risks becoming obsolete or irrelevant outside the Hopper ecosystem. Suggestion: Clarify how general the design is, and whether a fallback mode or alternative backend is possible.
2. No discussion on compiler or framework support
Although the paper presents an elegant low-level optimization, there is no clear path to integrating it into real-world ML compilers (like TensorRT, TVM, or PyTorch). Discuss whether a practical implementation could be exposed via kernel templates or integration with tools like Triton or CUTLASS.
3. Limited task and batch settings
All experiments are conducted with batch size = 1 and on specific LLMs (Llama2-7B, DeepSeek-V2-Lite). It’s unclear whether the speedups hold in multi-batch, mixed-length, or multitask environments, which are common in actual deployments. Suggestion: Discuss potential performance changes under more realistic workloads.
4. Potential for energy-performance tradeoffs is not explored
While the work optimizes latency, it does not address power or energy use, which is critical in datacenter-scale inference. Suggestion: Even if not measured, a brief comment on energy efficiency (e.g., due to reduced off-chip memory traffic) would strengthen the story.

**Quality:**

3

**Strengths And Weaknesses:**

Strengths

1. Originality: Proposing structured, cluster-level primitives for intra-block GPU communication is non-trivial and novel; Extending fusion beyond single kernels via coordinated scheduling of communication and computation is a clear step forward for system-level optimization in LLM serving.
2. Significance: This is not just a toy speedup: the ~1.6× end-to-end decoding speedup is meaningful and relevant to real-world LLM deployment at scale; The ideas are hardware-aware and reflect a strong grasp of actual GPU architecture, making them useful beyond just academic interest.
3. Clarity: The paper is mostly clear and readable. Diagrams (especially Fig. 6 and 7) help explain the concepts well; The structure (problem → primitives → fused dataflow → results) is logical and easy to follow.

Weaknesses

1. Over-reliance on NVIDIA Hopper (H100):
The entire framework depends heavily on low-level Hopper-only features (e.g., DSMEM, thread block clusters).
 The generalization to other GPUs (e.g., A100, non-NVIDIA) is not discussed at all. This makes the solution less broadly applicable, and the paper may quickly become obsolete as hardware evolves.
2. Fusion scope is still bounded by hardware (16-thread-block limit):
While the paper claims to "expand" fusion, it’s still limited by the current hardware cluster size.
There is no strong roadmap for scaling this beyond what Hopper already allows.
3. The method is system-level but lacks co-design with compiler/runtime stack:
There is no integration or discussion of compiler-level scheduling, which is critical for making this approach portable or usable at scale. How would a user actually integrate this into their LLM stack? Would HuggingFace or TensorRT need to be rewritten?
4. Ablation and analysis are shallow in parts:
The ablation on DSMEM (e.g., Fig. 13) is helpful but too limited. What happens to performance if you scale to batch sizes >1, or mix prompt lengths? A stronger analysis of latency breakdowns, kernel utilization, or memory pressure would help explain where the 1.6× speedup comes from in more detail.

---

> ### Author Rebuttal · Authors · 2025-07-30
>
> Thank you for your thoughtful feedback and suggestions on our paper. We would like to provide more clarification.
>
> To answer your concerns:
> 1. **Generality on other hardwares.** The techniques and rationale implemented in ClusterFusion are not limited to NVIDIA Hopper GPUs. Our fundamental goal is to enable holistic dataflow design at the larger scale. As discussed in the background section, existing fragmented design makes it difficult to achieve globally optimal. We construct ClusterReduce and ClusterGather that enable on-chip cooperation between thread blocks. On A100 GPUs, where DSMEM is not available, our on-chip collective primitives are not supported. However, as shown in Section 4.3 and Figure 13, we implement ClusterReduce and ClusterGather using global memory as a fallback, and still achieve competitive performance due to reduced kernel launches and improved holistic dataflow design. Below are the TPOT (ms) of Llama2-7B on A100:
>
> | Seqlen\Framework | SGLang | vLLM | ClusterFusion |
> |------|------|------|------|
> | 4k | 16.8   | 16.5 | **14.1** |
> | 8k | 17.1   | 16.9 | **14.9** |
>
> Moreover, DSMEM-like inter-core communication is becoming a mainstream direction in modern architectures[1,2,3,4]. These trends indicate that ClusterFusion’s primitives and dataflow strategies are compatible with emerging hardware. Also, the primitives and holistic dataflow can be mapped to other platforms. On AMD GPUs[1], SMs, shared memory, and DSMEM used in our algorithm can be naturally mapped to compute units, local data share (LDS), and LDS-to-LDS interconnects (e.g., in RDNA 4). On dataflow architectures such as Graphcore IPUs[3] and Cerebras Wafer-Scale Engines[4], our fine-grained task partitioning strategy remains applicable across the many-core compute fabric of these systems. The holistic dataflow design philosophy as well as our inter-core tree-pattern ClusterReduce and ClusterGather can be further explored on these platforms to fully exploit their on-chip interconnects and distributed memory hierarchy. We are actively collaborating with engineers to implement our primitives on other GPU like MI300X/MI308X GPUs, enabling the same holistic, end-to-end dataflow optimization.
>
> 2. **Integration into frameworks/compilers.** We have already integrated our fused kernels into the PyTorch stack via pybind. This is foundation for future integration into real production frameworks such as SGLang, vLLM, and TensorRT-LLM. For the evaluation shown in our paper, we have already use our wrapped PyTorch-compatible interfaces. These interfaces are aligned with existing kernel stack and support drop-in replacement. For framework integration, we need to replace the original kernels used in other frameworks with our ClusterFusion kernel but no CUDA runtime modifications. For CUTLASS integration, we can append our templates of ClusterReduce/ClusterGather primitives into CUTLASS template library. For ML compilers integration, we need to add a cluster pass which automatically group some thread blocks into a thread block cluster according to the data dependency for reduce/gather.
> 3. **About FFN, RMSNorm and RoPE.** We have extended our primitives to fuse other modules like FFN, and the corresponding implementation(codes) is included in our supplementary material. In short, FFN fusion is feasible, but the performance gain is limited.
> Firstly, the FFN kernel must be separated from the Output projection, because the output projection in Attention introduces a all-to-one data dependency: all thread blocks must reduce their output to the same address. This is not feasible to resolve fully on-chip via DSMEM without falling back to global memory, as the limited cluster size. Secondly, we designed a fused FFN kernel where we partition the intermediate size across blocks within a cluster, and split the output dimension across different clusters. ClusterReduce is then used to compute partial sums across blocks on chip, followed by local element-wise SiLU and residual addition. However, due to the large intermediate dimensions in most mainstream LLMs (e.g., 11008 or higher), each thread block must handle a huge reduction payload. As shown in our profiling results (Section 2.3 and Figure 5), the limited DSMEM bandwidth in this case prevents us from gaining significant on-chip communication benefits. However, FFN fusion still helps reduce kernel launch overhead and global memory traffic and achieves comparable latency to existing frameworks:
>
> | Framework         | FFN Latency (us) |
> |-------------------|------------------|
> | SGLang            | 115              |
> | vLLM              | 113              |
> | TensorRT-LLM      | 107              |
> | MLC-LLM           | 110              |
> | Ours  | **118**        |
>
> Regarding RMSNorm and RoPE, these components are omitted from the paper for clarity, but are indeed included in our fused kernels. Since they can be naturally fused using shared memory or registers without requiring ClusterReduce or ClusterGather primitives. For simplicity, we did not illustrate them in the figures, but their full implementations are provided in the supplementary material.
>
> 4. **The performance of multi-batch and variable length.** ClusterFusion delivers speedup across both batch size = 1 and batch size > 1 scenarios. For batch size > 1, we provide empirical results in the Appendix of the supplementary material, where we observe an average end-to-end speedup of 1.2×. As for variable-length prompts, supporting fully dynamic batching requires extending the attention module to handle runtime index computation and layout alignment. This extension is technically straightforward and will not compromise the performance benefits of our fused design, since the core communication and computation patterns remain unchanged. We are actively working on this enhancement extending our implementation to better support these use cases.
> 5. **Tuner usage.** We expose cluster_size as a tunable parameter for users and offline auto-tuning. In our current implementation, cluster_size is a template parameter at the kernel level. Based on our profiling (Section 2.3), larger values like 16 are generally discouraged. So, we provide a simple tuner that performs offline benchmarking over a small candidate set (e.g., 2, 4, 8). The tuner selects the best-performing configuration for a given input models.
> 6. **Energy optimizations.** Thanks for your suggention about discussion of energy-performance tradeoff. In section 4.2 and Figure 12, we report the reduced off-chip memory traffic dut to ClusterFusion optimizaions. These changes naturally reduce the frequency of global memory accesses and kernel launches, both of which are known to contribute to energy consumption on modern GPUs. We will include a discussion of these implications in the final version.
>
> [1] AMD "RDNA4" Instruction Set Architecture.
>
> [2] NVIDIA Blackwell Architecture Technical Brief.
>
> [2] Scaling Deep Learning Computation over the Inter-core Connected Intelligence Processor with T10, SOSP 2024.
>
> [3] WaferLLM: Large Language Model Inference at Wafer Scale, OSDI 2025.

---

### Official Review · Reviewer_q92t · 2025-07-06

**Clarity:** 4
**Significance:** 2
**Originality:** 2
**Rating:** 4
**Confidence:** 4

**Summary:**

The paper addresses decoding latency in LLM inference caused by fragmented kernels and frequent off-chip memory sync. Although NVIDIA Hopper GPUs offer low-level DSMEM, they lack high-level communication primitives.

**ClusterFusion** fills this gap by introducing two on-chip collective operations:
- **ClusterReduce**
- **ClusterGather**

These enable QKV, attention, and output projections to remain on-chip, cutting off-chip transfers and boosting performance. On H100 GPUs with Llama2-7B and DeepSeek-V2-Lite, ClusterFusion achieves a **1.61×** speedup over SGLang, vLLM, TensorRT-LLM, and MLC-LLM.

The paper contributes:
1. **Profiling** of decoding dataflow inefficiencies.
2. **High-level primitives** for structured inter-block communication on Hopper.
3. **ClusterFusion framework** that expands operator fusion and reduces inference latency.

**Questions:**

1. **Cluster Size (k) Selection**
   - **Issue:** The paper lacks concrete guidance on choosing the cluster size \(k\).
   - **Request:** Provide an analytical or discussion for \(k\) selection and for figure 11, could you deliver a comprehensive for more different models (e.g., reasoning model)

2. **Experiment Clarify**
   - **Issue:** It's unclear for FlashInfer [45], Triton [40] and TVM usage.
   - **Request:** Could you clearly tell me for vLLM, SGlang, etc, what's the actual kernel we are using for each framework. For example, do we use triton-based kernel or we do use torch.compile generated kernel?

3. **FFN analysis**
   - **Issue** I am confused why author also use these two primitives for MLP.
   - **Request** Could you give ma an analysis or discussion for using these two primitives for MLP? And what's the potential benefits?

4. **Evaluation on Long Contexts and Scalability to More Models**
   - **Issue:** Claims of benefits for long-sequence decoding are not validated on truly large contexts and larger / multi-modality model.
   - **Request:** Report latency and accuracy on models handling > 100 K token contexts (e.g., DeepSeek R1, Qwen-Long, Llama-4) to substantiate your scalability claims.
    - **Request:** Provide empirical results or detailed performance projections for multi modality or larger model (e.g., llama 3.3 70 B), and discuss any emerging bottlenecks or diminishing returns.

5. **Cross-Architecture Portability**
   - **Issue:** The approach is demonstrated only on NVIDIA Hopper GPUs.
   - **Request:** Discuss architectural assumptions and outline how ClusterReduce/ClusterGather could be implemented on other platforms (e.g., AMD/Intel GPUs or TPUs). If possible, include a small-scale result on one alternative accelerator.

**Ethical Concerns:**

["NO or VERY MINOR ethics concerns only"]

**Final Justification:**

The technical concerns I had with this paper are mostly resolved. I will raise my point.

But I hope to see the final manuscript include our discussion in the full paper/appendix and include the points we discussed during rebuttal.

**Limitations:**

Yes. There is no potential negative societal impact of this work.

**Paper Formatting Concerns:**

None observed. Paper adheres closely to the NeurIPS 2025 guidelines.

**Quality:**

4

**Strengths And Weaknesses:**

## Strengths

1. **Quality**
   - The paper delivers a rigorous motivation analysis to algorithm design. I do love the diagrams, pseudocode and smooth storyline of paper that walk me through on-chip end2end design.
   - Empirical measurements are comprehensive—covering per-kernel breakdowns, DSMEM bandwidth vs. global memory, and SM-to-SM link utilization.

2. **Clarity**
   - **ClusterReduce** and **ClusterGather** are crisply specified: their APIs map directly to common data-parallel patterns (all-reduce, broadcast) and hide DSMEM complexities.

3. **Significance of Results**
   - The approach first proposes such primitive and directly targets production-critical workloads. On real hardware (NVIDIA H100), ClusterFusion yields a consistent **1.61×** end-to-end speedup over four competing inference frameworks.

---

## Weaknesses

1. **Originality vs. Engineering Effort**
   - The primitives rest on NVIDIA PTX’s DSMEM features; while the abstraction is valuable, it does not introduce a fundamentally new communication algorithm or theoretical model. The novelty is primarily in the high-level API and scheduler integration rather than in core algorithms.

2. **Generality Across Architectures**
   - The paper mentioned different architecture may suffer from these issues, but all experiments target Hopper GPUs. It remains unclear how ClusterReduce/Gather would perform—or even map—to architectures without hardware DSMEM (e.g., AMD ROCm, TPUs). A discussion or prototype on alternate hardware would bolster claims of broad applicability.

3. **Scalability to Very Large Models and Contexts**
   - Evaluation caps at 7 B-parameter models and 16 K token contexts. It is left open whether on-chip buffers suffice for, say, 70 B models or 100 K-token sequences without falling back to global memory, which would erode benefits.

4. **Depth of Algorithmic Detail**
   - The paper’s pseudocode for collective setup omits corner-case handling (e.g., SM load imbalance, DSMEM bank conflicts). A deeper dive into these implementation subtleties would aid reproducibility.
   - Latency models for DSMEM vs. global memory are presented qualitatively; a formal performance model (e.g., roofline-style) could clarify when the method breaks even.

5. **Clarity in Presentation of Limitations**
   - While the authors briefly note DSMEM capacity limits, they do not quantify how often fusion must be “split” across multiple passes or how that impacts latency.

---

> ### Author Rebuttal · Authors · 2025-07-30
>
> Thank you for your thoughtful feedback on our paper. We would like to provide more clarification.
>
> To answer your concerns:
> 1. **Cluster Size (k) Selection.** Based on our profiling results in Section 2.3 and Figure 5, we observe that using an excessively large cluster size can degrade performance due to reduced effective SM bandwidth and lower active SM. To mitigate this, we recommend choosing a cluster size less than or equal to 8 for most workloads. Additionally, to facilitate data tiling and ensure alignment between data layout and thread-block mapping, we suggest using even cluster sizes such as 2, 4, or 8—resulting in a small but practical space. We also expose cluster_size as a tunable parameter in our implementation, since its optimal value is tightly coupled with several factors such as tile size and kernel-level parallelism. In practice, this introduces a trade-off: when the cluster count is large and each cluster handles a small amount of data, a smaller cluster size leads to sufficient parallelism and more favorable tiling. On the other hand, when each cluster’s workload is relatively large, a larger cluster size is needed to efficiently amortize inter-block communication. This behavior is reflected in our results in Section 4.1 and Figure 11, where smaller cluster sizes (2 or 4) are preferred for many-head attention, while larger ones become more favorable as the head count decreases or head dimension grows.
> 2. **Kernels used in each framework.** In our experiments, we adopt the default operators used by each framework under their officially recommended configuration[4,5,6,7], with `torch.compile` enabled to utilize optimized Triton-based kernels for decoding. The following table summarizes the actual attention and GEMM kernels used in each baseline framework:
>
> | Framework      | Attention Kernel                                    | QKV/FFN GEMM Kernel                    |
> |----------------|-----------------------------------------------------|----------------------------------------|
> | **SGLang**     | FlashInfer (`BatchDecodeWithPagedKVCacheWrapper`)  | Torch.compile generated Triton Kernel |
> | **vLLM**       | FlashAttention-3 (`flash_attn_with_kvcache`)        | Torch.compile generated Triton Kernel|
> | **TensorRT-LLM** | FlashAttention-3 (`flash_attn_with_kvcache`)      | Torch.compile generated Triton Kernel  |
> | **MLC-LLM**    | TVM-generated kernel                                | TVM-generated kernel                  |
>
> All of these kernels represent SOTA implementations in their respective frameworks. Our method surpasses their performance, demonstrating both the generality and effectiveness of our approach. We will include detailed information about these kernel in the final version of the paper.
>
> 3. **About FFN.** We have extended our primitives to fuse other modules like FFN, and the corresponding implementation(codes) is included in our supplementary material. In short, FFN fusion is feasible, but the performance gain is limited.
> Firstly, the FFN kernel must be separated from the Output projection, because the output projection in Attention introduces a all-to-one data dependency: all thread blocks must reduce their output to the same address. This is not feasible to resolve fully on-chip via DSMEM without falling back to global memory, as the limited cluster size. Secondly, we designed a fused FFN kernel where we partition the intermediate size across blocks within a cluster, and split the output dimension across different clusters. ClusterReduce is then used to compute partial sums across blocks on chip, followed by local element-wise SiLU and residual addition. However, due to the large intermediate dimensions in most mainstream LLMs (e.g., 11008 or higher), each thread block must handle a huge reduction payload. As shown in our profiling results (Section 2.3 and Figure 5), the limited DSMEM bandwidth in this case prevents us from gaining significant on-chip communication benefits. However, FFN fusion still helps reduce kernel launch overhead and global memory traffic and achieves comparable latency to existing frameworks:
>
> | Framework         | FFN Latency (us) |
> |-------------------|------------------|
> | SGLang            | 115  |
> | vLLM              | 113|
> | TensorRT-LLM      | 107 |
> | MLC-LLM           | 110|
> | Ours  | **118** |
>
> 4. **Evaluation on longer context and larger models.** First, regarding long-context capability, we have extended our evaluation of Llama2-7B to >100K context length. In this setting, the TPOT(ms) of our fused kernel against baselines is as follows:
>
> | Seqlen\Framework | SGLang | vLLM | Ours |
> | ------ | ------ | ------ | ------ |
> | 128k   | 33.5   | 33.7 | **25.6** |
>
> Now, we have already evaluated a range of model sizes under common single-GPU inference settings. As for scalability to larger and multi-modal models with long context that require distributed inference, we have confirmed that our kernel maintain measurable per-device speedups even when integrated into a multi-GPU inference pipeline. As an initial step, we tested a distributed setup for DeepSeek-R1-671B with TP=16, where our single-GPU fusion kernels still yield measurable gains in the per-device computation stage. In this setting, the TPOT(ms) of our fused kernel against baselines is as follows:
>
> | Seqlen\Framework | SGLang | vLLM | Ours     |
> | ------ | ------ | ------ | ------ |
> | 128k   | 38.1   | 38.5 | **37.1** |
>
> While these results are currently limited to per-device computation time, they demonstrate the effectiveness of our approach for integration into multi-GPU/node inference. When scaling to larger models such as DeepSeek-R1, the speedup diminishes slightly (e.g., from 1.3× on LLaMA2-7B to ~1.1× on R1). This is because operator fusion has less room to eliminate overheads in larger models. Our future work focus on the extensions that co-optimize communication and memory access which will further improve performance. To address cross-device scalability, our implementation allows seamless insertion of collective communication operations (e.g., ring-based all-reduce, gather) without disrupting kernel scheduling. This enables future support for end-to-end fusion across both intra- and inter-GPU workloads with minimal integration effort.
>
> 5. **Generality on other hardwares.** The techniques and rationale implemented in ClusterFusion are not limited to NVIDIA Hopper GPUs. Our fundamental goal is to enable holistic dataflow design at the larger scale. As discussed in the background section, existing fragmented design makes it difficult to achieve globally optimal. We construct ClusterReduce and ClusterGather that enable on-chip cooperation between thread blocks. The primitives can be mapped to other platforms. On AMD GPUs, SMs, shared memory, and DSMEM used in our algorithm can be naturally mapped to compute units, local data share (LDS), and LDS-to-LDS interconnects (e.g., in RDNA 4[1]). On dataflow architectures such as Graphcore IPUs [2] and Cerebras Wafer-Scale Engines [3], our fine-grained task partitioning strategy remains applicable across the many-core compute fabric of these systems. The holistic dataflow design philosophy as well as our inter-core tree-pattern ClusterReduce and ClusterGather can be further explored on these platforms to fully exploit their on-chip interconnects and distributed memory hierarchy. We are actively collaborating with engineers to implement our primitives on other GPU like MI300X/MI308X GPUs, enabling the same holistic, end-to-end dataflow optimization.
> 6. **About novelty.** While reduce and gather are common primitives in distributed system, they are rarely considered in chip-scale with interconnect. This enable new optimization scope and opportunities, and our novelty mainly lies in the exploration of on-chip dataflow via interconnect. On Hopper GPUs, blocks within a cluster are fully connected via DSMEM, as illustrated in Figure 4 and Figure 6. And as shown in Section 2.3 and Figure 5, DSMEM performance is non-trivial and not automatically efficient—naïve use often underperforms. Our primitives apply structured, tree-pattern, and collective scheduling to fully exploit DSMEM, enabling practical performance gains in real LLM workloads.
> 7. **Clarity in Presentation of Limitations.** As mentioned in **3. About FFN.**, our on-chip fusion is "split" when there are large-scale data dependencies such as all-to-one reductions that cannot be fully handled within DSMEM due to capacity constraints. In these cases, our implementation falls back to global memory synchronization. However, such cases are rare in typical LLM inference workloads and the same data dependency patterns would also cause global memory traffic in baseline frameworks. Therefore, our method still maintains a performance advantage. We will make this behavior more explicit in the final version.
> 8. **Depth of Algorithmic Detail.** We appreciate the reviewer’s suggestion. While our current pseudocode omits certain corner cases, our implementation can readily accommodate such scenarios. For example, load balancing can be enhanced by adaptive task-to-SM mapping in the start of the kernel, and DSMEM access patterns can be tuned in our primitive template to avoid bank conflicts. These extensions can be integrated with minimal change, thanks to the modular and general nature of our collective abstraction. As for formal performance model , We will include a roofline model that guide applicability across different workload sizes in the final version to improve clarity.
>
> [1] AMD "RDNA4" Instruction Set Architecture.
>
> [2] Scaling Deep Learning Computation over the Inter-core Connected Intelligence Processor with T10, SOSP 2024.
>
> [3] WaferLLM: Large Language Model Inference at Wafer Scale, OSDI 2025.
>
> [4] SGLang Blog, torchcompile-latency-optimizations, 2024-09-04.
>
> [5] vLLM Blog, torch.compile and Piecewise CUDA Graphs, 2025/01/27.
>
> [6] TensorRT-LLM Doc, Github.
>
> [7] MLC-LLM Doc, Github.

---

### Note · Authors · 2025-08-15

We sincerely thank the AC and reviewers for their time, effort, and constructive feedback throughout the review process. In the rebuttal, we provided additional clarifications, extended experiments across diverse configurations, and new analyses that further illustrate the generality of our method and quantify its benefits in realistic deployment scenarios. These include clarifying the design of ClusterReduce and ClusterGather primitives leveraging distributed shared memory on NVIDIA Hopper GPUs under different communication patterns, demonstrating their integration into the ClusterFusion framework for broader operator fusion in LLM decoding, and presenting additional evaluations showing average 1.61× end-to-end speedup over state-of-the-art inference frameworks.

We once again thank the AC for carefully overseeing the review process and considering our final clarifications, and we thank the reviewers for their thoughtful engagement, which has helped us present our work with greater clarity and completeness.

---

### Decision · Program_Chairs · 2025-09-17

**Decision:**

Accept (poster)

**Comment:**

This paper proposes ClusterFusion, a new system that accelerates LLM inference by fusing multiple steps—such as QKV projection, attention, and output projection—into a single optimized computation block. All reviewers agree that this work makes solid contributions, and the AC recommends acceptance.